# Provably Data-driven Lagrangian Relaxation for Mixed Integer Linear Programming

**Tung Quoc Le** [1]   **Anh Tuan Nguyen** [2]   **Viet Anh Nguyen** [3]

## Abstract

Lagrangian Relaxation (LR) is a powerful technique for solving large-scale mixed-integer linear programs, particularly those with decomposable structures, such as vehicle routing or unit commitment problems. By relaxing the coupling constraints, LR enables parallel subproblem solving and often yields tighter dual bounds than standard linear programming relaxations, which is crucial for efficient branch-and-bound pruning. While recent empirical work has shown promising results using machine learning to predict these multipliers, a theoretical understanding of such methods remains an open question. In this work, we bridge this gap by analyzing the problem of learning LR through the lens of data-driven algorithm design, i.e., a statistical learning problem over a distribution of problem instances. Our contributions are as follows: first, we derive a generalization bound of $\mathcal{O}(s^{1.5}/\sqrt{N})$ for the learned multipliers, where $s$ is the number of coupling constraints and $N$ is the sample size. Second, we provide a minimax lower-bound of $\Omega(s/\sqrt{N})$, proving that a linear dependency is unavoidable. Third, we constructively close this theoretical gap by proving that stochastic gradient ascent with averaging achieves the minimax optimal rate $\Theta(s/\sqrt{N})$. Finally, we extend our framework to the learning-to-warm-start setting, proving that it achieves a fast, minimax-optimal rate of $\Theta(s/N)$ and establishing a theoretical advantage over direct multiplier prediction.

[1]Université Grenoble Alpes, LJK, CNRS, Grenoble INP, 38000 Grenoble, France [2]Carnegie Mellon University, Machine Learning Department [3]Chinese University of Hong Kong, Department of Systems Engineering and Engineering Management. Correspondence to: Anh Tuan Nguyen <atnguyen@cs.cmu.edu>.

*Proceedings of the $43^{rd}$ International Conference on Machine Learning*, Seoul, South Korea. PMLR 306, 2026. Copyright 2026 by the author(s).

## 1. Introduction

Mixed Integer Linear Programming (MILP) (Conforti et al., 2014; Wolsey, 2020) is one of the most fundamental problems in optimization literature that plays a critical role in many industrial areas, including supply chains and logistics (Laporte, 1992), energy and power systems (Carrión & Arroyo, 2006), and finance (Grossmann, 2005), among others. While modern solvers (e.g., branch-and-cut (Mitchell, 2002)) have made tremendous advances, solving large-scale MILP instances to optimality remains a computationally intensive task due to the combinatorial explosion of the search tree. This necessitates advanced decomposition techniques to accelerate the solving process.

For many real-world MILP problems, including the Vehicle Routing Problem (VRP) (Toth & Vigo, 2014) and the Unit Commitment Problem (Saravanan et al., 2013), the complexity arises from a small set of *coupling* constraints (e.g., shared resources or capacity limits) linking together otherwise independent sub-problems. Lagrangian relaxation (LR) (Fisher, 1981) addresses this issue by dualizing these coupling constraints into the objective function. Concretely, let $P = (c, A, b, C, d)$ be a MILP problem instance of the form

$$\text{OPT}(P) \triangleq \begin{cases} \min & c^\top x \\ \text{s.t.} & x \in \mathbb{R}_+^m \times \{0,1\}^p, \ Ax \geq b, \ Cx \geq d, \end{cases}$$

where $Ax \geq b$ represents the coupling constraints. Using $\pi \geq 0$ as the dual variable, LR dualizes the coupling constraints into the objective function as follows

$$u(\pi, P) \triangleq \begin{cases} \min & c^\top x + \pi^\top (b - Ax) \\ \text{s.t.} & x \in \mathbb{R}_+^m \times \{0,1\}^p, \ Cx \geq d. \end{cases}$$

LR offers two important advantages for MILP solvers. First, it decomposes the problem into smaller, often tractable sub-problems that can be solved in parallel; see Appendix A for an example of such decomposition in VRP. Second, because $u(\pi, P) \leq \text{OPT}(P)$ and because these subproblems retain their integrity constraints, LR yields a better, often significantly tighter, objective lower-bound than the continuous relaxation (CR) (Geoffrion, 2009). These tighter bounds are instrumental in pruning the branch-and-bound tree and massively accelerating exact MILP solvers.

However, the efficiency of LR relies heavily on how fast we can identify the optimal values of the Lagrangian multipliers $\pi$. Given a problem instance $P$, finding the optimal multipliers that maximize the dual bound, and thus provide the best pruning power, is a non-smooth concave optimization problem. Consequently, the computation cost of finding good multipliers may sometimes outweigh the benefits of a tighter bound.

In many practical settings, the optimization problem instances $P$ are not isolated but appear as related samples coming from an application-specific problem distribution $\mathcal{D}$. For example, in the VRP case, we may observe daily routing demands on the same road network. Based on this fact, recent *empirical* works (Demelas et al., 2024) proposed learning the Lagrangian multipliers $\pi^*(\mathcal{D})$ for such problem distribution based on the past problem instances. Given a new test problem instance arriving from the same problem distribution, the learned $\pi^*(\mathcal{D})$ can be used to either (1) warm-start the sub-gradient method and drastically reduce the number of iterations required to reach convergence, or (2) approximate the dual bound directly without further re-calculation. However, despite some promising empirical results, this learning-augmented approach currently lacks a rigorous theoretical foundation.

**Contributions.** In this work, we lay the theoretical foundation for Lagrange multiplier learning by analyzing LR through the lens of data-driven algorithm design (Gupta & Roughgarden, 2020; Balcan, 2020). Concretely, we formalize learning the Lagrangian multipliers as a statistical learning problem and provide the first rigorous sample-complexity guarantee for this task. Our contributions are fourfold:

1. First, we derive a generalization guarantee upper-bound of $\mathcal{O}(s^{1.5}/\sqrt{N})$ for learning the Lagrangian multipliers that maximize the dual objective, where $s$ is the number of coupling constraints, and $N$ is the number of training problem instances.

2. Second, we establish a generalization minimax lower-bound of $\Omega(s/\sqrt{N})$, demonstrating that a linear dependence on the number of coupling constraints is unavoidable.

3. Third, we constructively close the resulting $\mathcal{O}(\sqrt{s})$ gap by analyzing the Stochastic Gradient Ascent (SGA) with the averaging algorithm, proving that it achieves the minimax optimal rate of $\mathcal{O}(s/\sqrt{N})$.

4. Finally, we extend our analysis to the learning-to-warm-start setting. We prove that this formulation admits a fast, minimax-optimal generalization rate of $\Theta(s/N)$, establishing a provable difference in sample complexity compared to direct multiplier prediction.

**Technical challenges and overview.** To establish the upper and lower bounds for generalization guarantees, we analyze the function class $\mathcal{U} = \{u_\pi : \mathcal{P} \to [-H, H] \mid \pi \in \mathbb{R}_+^s\}$, where $u_\pi(P) \triangleq u(\pi, P)$ and $H \in \mathbb{R}_+$ is a range bound of the function class. Unlike standard problems in learning theory literature, the structure of the function $u_\pi(P)$ is more complicated because $u_\pi(P)$ is inherently defined by an optimization problem. To overcome this challenge, we take the *dual view* perspective from data-driven algorithm design literature (Balcan, 2020): we will analyze the function $u_P(\pi) \triangleq u(\pi, P)$ acquired by fixing the MILP problem instance $P$ and treating the Lagrangian multipliers $\pi$ as variables. In doing so, we can exploit the favorable structure of the function class in Section 5.1 and then establish an upper-bound for the generalization guarantee for $\mathcal{U}$. For the lower bound, we employ a standard reduction in theoretical statistics from estimation to testing, using Fano's method from Theorem 3.3. To do this, we design a family of *hard* problem distributions distinguished by specific perturbations, controlled by a binary vector $v \in \{0, 1\}^s$, in their objective coefficient $c$. By constructing a hypercube packing of the controlling vector $v$, we induce a family of hard problem distributions $\mathcal{D}_v \in \Delta(\mathcal{P})$ that remain statistically hard to distinguish while their optimal multipliers $\pi^*(\mathcal{D}_v)$ remain geometrically separated, thereby establishing an information-theoretic lower-bound for the problem of data-driven Lagrangian relaxation learning.

**Outline.** We structure our paper as follows: Section 2 discuss the existing literature, while Section 3 collects key technical preliminaries. In Section 4, we present the setup of the Lagrangian relaxation learning framework. We present our main results on generalization guarantees, the minimax lower bound, and the optimal SGA algorithm in Section 5, while Section 6 extends these results to the learning-to-warm-start LR setting.

## 2. Literature Review

**Lagrangian relaxation (LR)** is a cornerstone technique for solving large-scale MILP problems, particularly those with decomposable structures (Fisher, 1981; Lemarechal, 2001). Unlike standard continuous relaxation that often yields loose bounds, LR maintains integrality constraints of sub-problems, providing tighter dual bounds (Geoffrion, 2009) that are crucial for pruning branch-and-bound/cut trees. However, the efficacy of LR hinges on finding good Lagrangian multipliers, often obtained via classical, computationally expensive methods such as subgradient ascent. This computational bottleneck motivates the use of learning-based approaches to predict high-quality multipliers (Demelas et al., 2024). Despite promising empirical results, this method lacks a theoretical foundation, which motivates our work.

**Learning to optimize (L2O)** is an active topic in machine learning (Van Hentenryck & Dalmeijer, 2024; Chen et al., 2022). Existing approaches L2O mostly fall into two categories: (1) end-to-end learning (i.e., amortized optimization) (Amos, 2023; Kool et al., 2018; Cappart et al., 2023), where models attempt to approximate the optimal solution directly, often *lacking feasibility and optimality guarantees*; and (2) solver-integrated learning, which configures specific components of exact solvers, such as branching strategies or cutting planes, to accelerate the solving process while naturally *maintaining feasibility and optimality* (Balcan et al., 2021b; 2022; Cheng & Basu, 2025; Nguyen & Nguyen, 2026). Predicting Lagrangian multipliers falls into the second category and has been investigated empirically (Demelas et al., 2024). Our paper aims to establish a theoretical foundation for this approach.

**Data-driven algorithm design** is a closely related research direction of L2O. Instead of worst-case analysis, it proposes adapting algorithms by tuning their internal parameters or components to a specific problem domain using historical problem instances from the problem distribution (Gupta & Roughgarden, 2020; Balcan, 2020). Data-driven algorithm design is an active line of work in both empirical validations and theoretical analyses across various domains, including sketching and low-rank approximation (Indyk et al., 2019; Li et al., 2023), tuning regularization parameters in regression models (Balcan et al., 2023), mixed integer linear programming (Balcan et al., 2022; 2018; Cheng & Basu, 2025), as well as various other generalization frameworks for analyzing their statistical guarantees (Balcan et al., 2021a; Bartlett et al., 2022; Balcan et al., 2025b;a; Le et al., 2026). Data-driven Lagrangian relaxation can be viewed as a specific instance of data-driven algorithm design.

# 3. Preliminaries

In this section, we will recall the necessary background and formalize the problem of data-driven learning Lagrangian relaxation.

## 3.1. Uniform Convergence and Rademacher Complexity

To quantify the learning-theoretic complexity of a real-valued function class, we use the Rademacher complexity (Bartlett & Mendelson, 2002).

**Definition 3.1** (Rademacher complexity, Bartlett & Mendelson (2002))**.** Let $\mathcal{U}$ be a real-valued utility function class of which each function takes inputs from the domain $\mathcal{P}$. Given a fixed set of samples $S = \{P_1, \ldots, P_N\} \subset \mathcal{P}$, the *empirical Rademacher complexity* of $\mathcal{U}$ with respect to the set $S$ is defined as

$$\hat{\mathscr{R}}_S(\mathcal{U}) = \frac{1}{N} \mathbb{E}_\sigma \left[ \sup_{u \in \mathcal{U}} \sum_{i=1}^{N} \sigma_i u(P_i) \right],$$

where $\sigma = (\sigma_1, \ldots, \sigma_N)$ are independent Rademacher variables. Given a distribution $\mathcal{D}$ over $\mathcal{P}$, the *Rademacher complexity* defined on $N$ inputs is defined as

$$\mathscr{R}_N(\mathcal{U}) = \mathbb{E}_{S \sim \mathcal{D}^N}[\hat{\mathscr{R}}_S(\mathcal{U})].$$

The next result connects Rademacher complexity to the expected generalization error.

**Theorem 3.2** (Uniform convergence via Rademacher complexity, Bartlett & Mendelson (2002))**.** *Let $\mathcal{D}$ be a distribution over the problem instance space $\mathcal{P}$, and let $S = \{P_1, \ldots, P_N\}$ be a set of $N$ i.i.d. samples drawn from $\mathcal{D}$. For any real-valued function class $\mathcal{U}$ that takes input in $\mathcal{P}$, the expected uniform deviation of the empirical mean from the true mean is bounded by*

$$\mathbb{E}_{S \sim \mathcal{D}^N} \left[ \sup_{u \in \mathcal{U}} \left( \mathbb{E}_{P \sim \mathcal{D}}[u(P)] - \frac{1}{N} \sum_{i=1}^{N} u(P_i) \right) \right] \le 2\mathscr{R}_N(\mathcal{U}).$$

## 3.2. Information-theoretic Minimax Lower Bound

Our approach establishes minimax lower bounds for any learning algorithm by reducing the estimation problem to multi-hypothesis testing using *Fano's method* (Wainwright, 2019). The idea is to construct a "hard" finite family of problem distributions $\mathcal{D}_{v \in V}$ indexed by $v$ so that

- The optimal parameters for different indices are far apart, that is, the parameters are separated by a distance at least $2\delta$ for some $\delta > 0$; and

- The distributions themselves are statistically hard to distinguish, that is, they have small Kullback-Leibler (KL) (Kullback & Leibler, 1951) divergence values.

If the distributions are too similar, no algorithm can reliably identify the correct index from a finite set of samples $S$. Therefore, the algorithm will frequently guess the wrong distribution, incurring an estimation error of at least $\delta$. Fano's method formalizes the idea as follows.

**Theorem 3.3** (Fano's method, Wainwright (2019))**.** *Let $\Pi$ be a parameter space and let $\{\mathcal{D}_v : v \in \Pi\}$ be a family of distributions. Let $\mathcal{V} = \{v^{(1)}, \ldots, v^{(M)}\} \subset \Pi$ be a $\delta$-packing set of $\Pi$, i.e., $\rho(v^{(i)}, v^{(j)}) \ge 2\delta$ for any $i, j \in \{1, \ldots, M\}$ and $i \ne j$. Let $J$ be a random index uniformly distributed over $\{1, \ldots, M\}$. Defining the Markov chain $J \to v_J \to S \to \pi$, where $S$ is a set of $N$ samples drawn from $\mathcal{D}_{v_J}$, then:*

$$\inf_\pi \sup_{v \in \mathcal{V}} \mathbb{E}\left[\rho(\pi, v)\right] \ge \delta \left( 1 - \frac{I(J; S) + \log 2}{\log M} \right),$$

*where $I(J; S)$ is the mutual information between the random index $J$ and the samples $S$, $\rho$ is a distance in $\Pi$, and $\pi : \mathcal{P}^N \to \Pi$ is an estimator.*

To apply Fano's method, a standard practice is to give an upper bound for the mutual information using the KL divergence. For $N$ i.i.d. samples, the mutual information is bounded by the pairwise KL divergence, i.e.,

$$I(J;S) \leq \frac{1}{M^2} \sum_{i,j=1}^{M} \mathrm{KL}(\mathcal{D}_{v^{(i)}}^{N} \| \mathcal{D}_{v^{(j)}}^{N})$$
$$\leq N \max_{i,j} \mathrm{KL}(\mathcal{D}_{v^{(i)}} \| \mathcal{D}_{v^{(j)}}). \tag{1}$$

## 4. Problem Settings

We formally define the Mixed Integer Linear Programming (MILP) setting and the corresponding statistical learning problem of data-driven Lagrangian relaxation.

**Setup.** We consider a general MILP instance $P = (c, A, b, C, d) \in \mathcal{P} \subset \mathbb{R}^{m+p} \times \mathbb{R}^{s \times (m+p)} \times \mathbb{R}^s \times \mathbb{R}^{t \times (m+p)} \times \mathbb{R}^t$ in the inequality constraints form:

$$\mathrm{OPT}(P) \triangleq \begin{cases} \min & c^\top x \\ \text{s.t.} & x \in \mathbb{R}_+^m \times \{0,1\}^p, Ax \geq b, Cx \geq d. \end{cases}$$

Here, $Ax \geq b$ represents the set of $s$ hard coupling constraints that make the problem computationally exhaustive. These constraints could be the linking constraints in Vehicle Routing Problem; see Appendix A for details.

By dualizing the $s$ coupling constraints with a Lagrangian multipliers vector $\pi \in \mathbb{R}_+^s$, we obtain the Lagrangian dual function $u(\pi, P)$:

$$u(\pi, P) \triangleq \begin{cases} \min & c^\top x + \pi^\top(b - Ax) \\ \text{s.t.} & x \in \mathbb{R}_+^m \times \{0,1\}^p, \ Cx \geq d. \end{cases}$$

It is a classical result that for any $\pi \in \mathbb{R}_+^s$, the value function $u(\pi, P)$ provides a valid lower bound on the optimal value objective, i.e., $u(\pi, P) \leq \mathrm{OPT}(P)$. Given a problem instance $P$, the best Lagrangian multipliers $\pi^*(P)$ corresponding to $P$ is found by solving the *Lagrangian dual problem*:

$$\pi^*(P) \in \arg\max_{\pi \in \mathbb{R}_+^s} u(\pi, P).$$

The data-driven settings assume that there is an application-specific problem distribution $\mathcal{D}$ over the set of problem instances $\mathcal{P}$. Optimally, we want to learn the best multipliers $\pi^*$ that maximize the expected Lagrangian value function over the problem distribution $\mathcal{D}$

$$\pi^*(\mathcal{D}) \in \arg\max_{\pi \in \mathbb{R}_+^s} \mathbb{E}_{P \sim \mathcal{D}}[u(\pi, P)].$$

Since the problem distribution $\mathcal{D}$ is unknown, the problem above is intractable. Instead, we observe a set of $N$ problem instances $S = \{P_1, \dots, P_N\} \sim \mathcal{D}^N$ and we consider the *empirical risk maximization* (ERM) maximizer $\hat{\pi}(S)$, where

$$\hat{\pi}(S) \in \arg\max_{\pi \in \mathbb{R}_+^s} \frac{1}{N} \sum_{i=1}^{N} u(\pi, P_i).$$

**Structural assumptions.** To ensure the tractability of this data-driven learning framework, we make the following standard regularity assumptions about the problem geometry and the search space throughout this work.

**Assumption 4.1** (Bounded constraint violation)**.** We assume there is a positive constant $B > 0$ such that with probability 1 for problem instances $P = (c, A, b, C, d)$ drawn from the problem distribution $\mathcal{D}$, for any feasible solution $x \in \mathcal{X} = \{x \in \mathbb{R}_+^m \times \{0,1\}^p \mid Cx \geq d\}$, the constraints violations are bounded coordinate-wise:

$$|b_k| \leq B \quad \text{and} \quad |(Ax)_k| \leq B, \quad \text{for all } k = 1, \dots, s.$$

**Assumption 4.2** (Restricted search domain)**.** We assume the learning algorithm searches for Lagrangian multipliers within a bounded hypercube $\Pi \subset \mathbb{R}_+^s$, defined as:

$$\Pi = \{\pi \in \mathbb{R}^s \mid 0 \leq \pi_k \leq \pi_{\max} \quad \forall k = 1, \dots, s\}$$

for some positive constant $\pi_{\max} > 0$.

Both assumptions are mild and commonly satisfied in practice. Assumption 4.1 naturally holds for almost all real-world MILP applications, such as in logistics and energy systems, where decision variables and parameters are bounded by physical constraints (e.g., truck capacities or electricity generator limits). Assumption 4.2 is a standard learning-theoretic requirement to ensure the compactness of the hypothesis class. Moreover, in practice, $\pi_{\max}$ can be chosen sufficiently large to include the optimal multipliers without loss of generality. Crucially, these assumptions naturally imply that the search space $\Pi$ has a bounded $\ell_2$-diameter $D = \pi_{\max}\sqrt{s}$, which is crucial for our further analyses.

**Objectives.** We aim to establish learning-theoretic guarantees for the performance of the learned multipliers $\hat{\pi}$. Specifically, we analyze the expected excess risk of the ERM minimizer $\hat{\pi}$, which measures the expected gap between the optimal dual bound and the dual bound achieved by the learned multipliers:

$$\mathcal{E}(\hat{\pi}) \triangleq$$
$$\mathbb{E}_{S \sim \mathcal{D}^N} \left[ \max_{\pi \in \Pi} \mathbb{E}_{P \sim \mathcal{D}}[u(\pi, P)] - \mathbb{E}_{P \sim \mathcal{D}}[u(\hat{\pi}(S), P)] \right].$$

To analyze this quantity, we use the concepts of *uniform convergence* introduced in Section 3. By a standard decomposition (Shalev-Shwartz & Ben-David, 2014), the expected risk can be upper-bounded by the *expected uniform deviation* of the function class $\mathcal{U} = \{P \mapsto u(\pi, P) \mid \pi \in \Pi\}$.

$$\mathcal{E}(\hat{\pi}) \leq$$
$$2\mathbb{E}_{S \sim \mathcal{D}^N} \left[ \sup_{\pi \in \Pi} \left( \mathbb{E}_{P \sim \mathcal{D}}[u(\pi, P)] - \frac{1}{N} \sum_{i=1}^{N} u(\pi, P_i) \right) \right].$$

Therefore, the main focus of this work is to analyze the expected uniform deviation.

# 5. Generalization Guarantee for Data-driven Learning Lagrangian Relaxation

In this section, we provide a generalization guarantee for data-driven learning for Lagrangian relaxation.

## 5.1. Geometric Properties

We first establish the geometric properties of the dual function $u(\pi, P)$, specifically its concavity and the boundedness of its subgradients. These properties are critical for bounding the complexity of the function class in subsequent sections.

**Proposition 5.1** (Concavity and subgradient). *Given any problem instance $P = (c, A, b, C, d) \in \mathcal{P}$, we have:*

(i) $u(\cdot, P)$ *is a concave function of $\pi$;*

(ii) $g(\pi, P) = b - Ax^*(\pi, P)$ *is a subgradient of $u(\pi, P)$ in the $\pi$ variable. Moreover, under Assumption 4.1, we have $\|g(\pi, P)\|_2 \leq 2B\sqrt{s}$.*

The proof of Proposition 5.1 follows the conventional arguments in convex analysis, and the full proof is relegated to Appendix C.2.

**Corollary 5.2** (Lipschitzness). *Given a problem instance $P$, we have $u(\cdot, P)$ is an $L$-Lipschitz function of $\pi$, where $L = 2B\sqrt{s}$. That is, for any $P \in \mathcal{P}$ and any $\pi, \pi' \in \Pi$, we have*

$$|u(\pi, P) - u(\pi', P)| \leq 2B\sqrt{s}\|\pi - \pi'\|_2.$$

*Proof.* This is a consequence of Proposition 5.1: the norm of a subgradient is *uniformly bounded* by $2B\sqrt{s}$. $\square$

Since $u(\cdot, P)$ is a concave function in $\pi$ for a given problem instance $P$, the vector $g(\pi, P)$ is technically the *super-gradient*, i.e., satisfying $u(\pi', P) \leq u(\pi, P) + g(\pi, P)^\top (\pi' - \pi)$ for any $\pi, \pi' \in \Pi$. However, to be consistent with standard literature on Lagrangian relaxation, which often generalizes the terminology of convex optimization, we refer to $g(\pi, P)$ as a *subgradient* of $u(\pi, P)$ throughout this work.

## 5.2. Rademacher Complexity Upper Bound

We now bound the Rademacher complexity of the function class $\mathcal{U} = \{P \mapsto u(\pi, P) : \pi \in \Pi\}$. To this end, we use $\mathcal{N}(\delta, \mathcal{U}, \|\cdot\|_{2,N})$ to denote the $\delta$-covering number of $\mathcal{U}$ with respect to the empirical $L_2$-norm (cf. Definition B.4).

**Lemma 5.3** (Covering number). *For any $\delta > 0$, we have*

$$\log \mathcal{N}(\delta, \mathcal{U}, \|\cdot\|_{2,N}) \leq s \log \left(1 + \frac{2B\pi_{\max}s}{\delta}\right).$$

For completeness, we present in Appendix B the related backgrounds on the $\delta$-covering properties. Next, we present the proof of Lemma 5.3.

*Proof of Lemma 5.3.* Given a problem instance $P$, the function $u(\cdot, P)$ is a $L$-Lipschitz function of $\pi$, where $L = 2B\sqrt{s}$ by Proposition 5.1. Therefore, an $(\delta/L)$-covering of the parameter space $\Pi$ in the $\ell_2$ metric induces an $\delta$-covering of the function class $\mathcal{U}$ in the empirical-$L_2$ metric. Then, a volume-metric apparent argument from Wainwright (2019, Lemma 5.7) shows that:

$$\log \mathcal{N}(\delta, \mathcal{U}, \|\cdot\|_{2,N}) \leq \log \mathcal{N}(\delta/L, \Pi, \|\cdot\|_2)$$
$$\leq s \log \left(1 + \frac{\text{diam}(\Pi)L}{\delta}\right).$$

Substituting $\text{diam}(\Pi) = \pi_{\max}\sqrt{s}$ from Assumption 4.2 and $L = 2B\sqrt{s}$ from Corollary 5.2, we have the final claim. $\square$

From Lemma 5.3, we can combine with Dudley's entropic integral (Lemma B.6) to establish an upper-bound for the Rademacher complexity of $\mathcal{U}$.

**Lemma 5.4** (Rademacher complexity upper-bound). *The Rademacher complexity of $\mathcal{U}$ is bounded by: $\mathscr{R}_N(\mathcal{U}) = \mathcal{O}\left(\frac{s^{1.5}}{\sqrt{N}}\right).$*

*Proof of Lemma 5.4.* Let $D = \text{diam}(\Pi)$ be the diameter of $\Pi$. We use Dudley's entropic integral (cf. Lemma B.6) and Lemma 5.3 to bound the empirical Rademacher complexity $\hat{\mathscr{R}}_S(\mathcal{U})$. Given a set $S$ of $N$ problem instances, we have

$$\hat{\mathscr{R}}_S(\mathcal{U}) \leq \inf_{\delta_0 > 0} \delta_0 + \int_{\delta_0}^{L \cdot D} \sqrt{\frac{\log \mathcal{N}(\delta, \mathcal{U}, \|\cdot\|_{2,N})}{N}} d\delta$$
$$\leq \inf_{\delta_0 > 0} \delta_0 + \sqrt{\frac{s}{N}} \int_{\delta_0}^{L \cdot D} \sqrt{\log(3LD/\delta)} d\delta.$$

Using the classic identity $\int_0^R \sqrt{\log(R/\delta)}d\delta = R \cdot \frac{\sqrt{\pi}}{2}$,[1] we have

$$\hat{\mathscr{R}}_S(\mathcal{U}) \lesssim \sqrt{\frac{s}{N}} \cdot \frac{3LD\sqrt{\pi}}{2}.$$

And finally, recall from Lemma 5.3, we have $L = 2B\sqrt{s}$ and $D = \pi_{\max}\sqrt{s}$, we conclude that

$$\hat{\mathscr{R}}_S(\mathcal{U}) = \mathcal{O}\left(\frac{s^{1.5}}{\sqrt{N}}\right),$$

where we treat $B$ and $\pi_{\max}$ as constants. Taking expectation over $S \sim \mathcal{D}^N$, we have the final conclusion. $\square$

We then have the following result, which is a consequence of Lemma 5.4 and Theorem 3.2.

---

[1] We abuse notation here to let $\pi$ denote the mathematical constant $\pi \approx 3.14$. In all other contexts, $\pi$ refers to the Lagrangian multipliers.

**Theorem 5.5** (Risk bound). *Let $S$ be a set of $N$ problem instance $P_1, \ldots, P_N$ drawn i.i.d. from a problem distribution $\mathcal{D}$. Then the expected excess risk of ERM minimizer $\hat{\pi}(S)$ is bounded by*

$$\mathcal{E}(\hat{\pi}) = \mathcal{O}\left(\frac{s^{1.5}}{\sqrt{N}}\right).$$

*Proof of Theorem 5.5.* Combining the definition of expected excess risk in Section 4, the generalization bound via Rademacher complexity in Theorem 3.2, and Lemma 5.4 above, we have the final conclusion. □

### 5.3. Lower Bound

Section 5.2 asserted that the expected excess risk of the ERM estimator decays at a rate of $\mathcal{O}(s^{1.5}/\sqrt{N})$. This naturally raises the question: can a more sophisticated learning algorithm achieve a better rate, perhaps one that is independent of the number of coupling constraints $s$? In this section, we give a *negative* answer to this question. We will show in Theorem 5.6 that there is no algorithm, regardless of its design, that can achieve an expected excess risk lower than $\Omega(s/\sqrt{N})$ in the worst case. This implies that the linear dependence on the number of constraints $s$ is intrinsic to the geometry of Lagrangian relaxation and cannot be overcome by better algorithm design.

**Theorem 5.6** (Minimax lower-bound). *Let $s \geq 16$ be the number of coupling constraints and $N$ be the sample size. For any learning algorithm $\pi$ that maps a dataset $S \sim \mathcal{D}^N$ to a Lagrangian multiplier $\pi(S)$, the minimax expected excess risk is lower-bounded by:*

$$\inf_{\pi} \sup_{\mathcal{D} \in \Delta(\mathcal{P})} \mathcal{E}(\pi) = \Omega\left(\frac{s}{\sqrt{N}}\right).$$

**Proof overview.** Our proof for Theorem 5.6 relies on a reduction from estimation to testing using Fano's method in Theorem 3.3. The core idea is to construct a discrete family of *hard* problem distributions $\{\mathcal{D}_v\}_{v \in \mathcal{V}}$, where $\mathcal{V}$ is a set of binary vectors with $\Omega(2^{s/8})$ elements, that are statistically distinguishable only with a large number of samples, yet whose optimal multipliers $\{\pi^*(\mathcal{D}_v)\}_{v \in \mathcal{V}}$ are geometrically distinct. The full proof can be found in Appendix C.3, we here sketch three key steps:

**Step 1: Construction of hard instances.** We first construct a family of distributions parameterized by a binary vector $v \in \{0, 1\}^s$, effectively reducing the learning problem to a high-dimensional parameter estimation problem. The following result establishes the general idea of that construction and its geometric property.

**Lemma 5.7** (Geometric separation). *There exists a set of $M \geq 2^{s/8}$ distributions $\{\mathcal{D}_{v^{(1)}}, \ldots, \mathcal{D}_{v^{(M)}}\}_{v^{(i)} \in \mathcal{V}}$ parameterized by an $s$-dimensional binary vectors $v^{(i)} \in$*
$\mathcal{V} \subset \{0, 1\}^s$, *such that for any distinct pair $v^{(i)} \neq v^{(j)}$, the $\ell_1$ distance between their optimal multipliers satisfies $\|\pi^*(\mathcal{D}_{v^{(i)}}) - \pi^*(\mathcal{D}_{v^{(j)}})\|_1 \geq \Omega(\sigma s)$, where $\sigma$ is some perturbation scale parameter.*

*Proof sketch of Lemma 5.7.* The proof relies on the construction of a family of distributions where identifying the optimal multiplier is equivalent to estimating high-dimensional binary vectors. This relies on the following steps:

- **Instance restriction**: We first restrict our problem instance $P$ to the form $(c, \mathbf{I}_s, \frac{1}{2} \cdot \mathbf{1}_s, 0_{t \times s}, 0_t)$

$$\min_{x \in \{0,1\}^s} c^\top x \quad \text{s.t.} \quad x_k \geq \frac{1}{2} \quad \forall k = 1, \ldots, s.$$

  For instances of this form, we show that the Lagrangian dual objective $u(\pi, P)$ can be written as $u(\pi, P) = \sum_{k=1}^{s} \min\left(\frac{\pi_k}{2}, c_k - \frac{\pi_k}{2}\right)$.

- **Distribution construction**: Since the constraints are fixed, the instance $P$ is determined solely by the objective vector $c$. This means we can define a problem distribution by constructing a distribution over the objective vector $c$ only. Given a binary vector $v \in \{0, 1\}^s$, inducing a problem distribution $\mathcal{D}_v$, we define a distribution on $c$ as follows: for any coordinate $k$, $c_k$ takes value in $\{\mu, \mu + \sigma\}$ for some $\mu, \sigma > 0$ and $\mu + \sigma < \pi_{\max}$, and (i) if $v_k = 1$, then $\mathbb{P}(c_k = \mu + \sigma) = \frac{1+\epsilon}{2}$, and $\mathbb{P}(c_k = \mu) = \frac{1-\epsilon}{2}$, (ii) if $v_k = 0$, then $\mathbb{P}(c_k = \mu + \sigma) = \frac{1-\epsilon}{2}$, and $\mathbb{P}(c_k = \mu) = \frac{1+\epsilon}{2}$. Here, $\epsilon > 0$ is a value we will choose later. Under this construction, we can show that $\pi^*(\mathcal{D}_v) = \mu \mathbf{1}_s + \sigma v$.

- **Distribution family construction**: Consequently, for two binary vector $v^{(i)}, v^{(j)} \in \{0, 1\}^s$, we have $\|\pi^*(\mathcal{D}_{v^{(i)}}) - \pi^*(\mathcal{D}_{v^{(j)}})\|_1 = \sigma \|v^{(i)} - v^{(j)}\|_1$. Applying Varshamov-Gilbert bound in Lemma C.1, we can select a subset of vectors $\mathcal{V} = \{v^{(1)}, \ldots, v^{(M)}\} \subset \{0, 1\}^s$ with pairwise Hamming distance $s/8$. This induced a family of problem distributions $\{\mathcal{D}_v\}_{v \in \mathcal{V}}$ that satisfies the geometric separation property.

This leads to the postulated claim. □

**Step 2: Statistical indistinguishability.** To apply Fano's inequality, we must demonstrate that the distributions are statistically hard to identify despite the large geometric separation of their optimal multipliers established in Step 1. We quantify this difficulty by upper-bounding the KL divergence between the distributions in our constructed family.

**Lemma 5.8.** *Let $\{\mathcal{D}_v\}_{v \in \mathcal{V}}$ be the family of distributions defined in Step 1 with bias parameters $\epsilon \in (0, 1/2)$. For any distinct pair of binary vectors $v, v' \in \mathcal{V}$, the KL divergence between the corresponding $N$-sample product measures is bounded by $\mathrm{KL}(\mathcal{D}_v^N \| \mathcal{D}_{v'}^N) \leq 4Ns\epsilon^2$.*

*Proof.* Here, we exploit the independent structure of the constructed distribution to decompose the KL divergence. First, by the additivity of KL divergence for product measures, we have $\mathrm{KL}(\mathcal{D}_v^N \parallel \mathcal{D}_{v'}^N) = N \cdot \mathrm{KL}(\mathcal{D}_v \parallel \mathcal{D}_{v'})$. Second, for any problem distribution $\mathcal{D}_v$, the distribution of the objective coefficient $c_k$ (determined by a binary value $v_k$) is independent due to our construction. Therefore, $\mathrm{KL}(\mathcal{D}_v \parallel \mathcal{D}_{v'}) = \sum_{k=1}^{s} \mathrm{KL}(\mathcal{D}_{v_k} \parallel \mathcal{D}_{v'_k})$, where $\mathcal{D}_{v_k}$ denotes the marginal distribution of $c_k$ induced by $v_k$. Finally, note that $\mathrm{KL}(\mathcal{D}_{v_k} \parallel \mathcal{D}_{v'_k})$ is equivalent to the KL divergence between two Bernoulli distributions with parameters $p = \frac{1+\epsilon}{2}$ and $q = \frac{1-\epsilon}{2}$. Using $\chi^2$-upper bound, we have $\mathrm{KL}(\mathcal{D}_{v_k} \| \mathcal{D}_{v'_k}) \leq \chi^2(\mathcal{D}_{v_k} \| \mathcal{D}_{v'_k}) = \frac{4\epsilon^2}{1-\epsilon^2} = \mathcal{O}(\epsilon^2)$ for $\epsilon \in (0, 1/2)$. Summing over at most $s$ different coordinates yields the final conclusion. $\square$

**Step 3: Risk to parameter estimation reduction.** Finally, we link the hardness of parameter estimation to generalization risk by analyzing the geometry of the Lagrangian dual value function. We show that the objective is locally sharp, meaning that any estimation in the $\ell_1$-norm translates to a linear penalty in the excess risk.

**Lemma 5.9.** *Let $\mathcal{D}_v$ be any distribution in the constructed family with bias parameter $\epsilon \in (0, 1/2)$. For any estimator $\pi$, the expected excess risk is lower-bounded by*

$$\mathbb{E}_{P \sim \mathcal{D}_v}[u(\pi^*(\mathcal{D}_v), P) - u(\pi, P)] \geq \frac{\epsilon}{2}\|\pi^*(\mathcal{D}_v) - \pi\|_1.$$

*Proof sketch.* We analyze the expected Lagrangian dual value function $R(\pi) \triangleq \mathbb{E}_{P \sim \mathcal{D}_v}[u(\pi, P)]$. Due to our construction where $u(\pi, P)$ separates coordinate-wise and the distribution of $c_k$ induced by $v_k$, we can show that $R(\pi)$ can be decomposed as $R(\pi) = \sum_{k=1}^{s} \mathcal{J}_k(\pi_k)$, where $\mathcal{J}_k(\pi_k) = \mathbb{E}_{c \sim \mathcal{D}_v}\left[\min\left(\frac{1}{2}\pi_k, c_k - \frac{\pi_k}{2}\right)\right]$.

We then analyze the (sub)gradient of the univariate function $\mathcal{J}_k$. Recall that $\mathcal{J}_k$ is a weighted average of two *tent* functions centered at $\mu$ and $\mu + \sigma$. If $v_k = 1$, explicit calculation gives us $(\pi^*(\mathcal{D}_v))_k = \mu + \sigma$ and we have the following case:

- If $\pi_k < \mu$, we show that $\mathcal{J}_k((\pi^*(\mathcal{D}_v))_k) - \mathcal{J}_k(\pi_k) = \frac{\epsilon}{2}((\pi^*(\mathcal{D}_v))_k - \mu) + \frac{1}{2}(\mu - \pi_k) \geq \frac{\epsilon}{2}|\pi_k - (\pi^*(\mathcal{D}_v))_k|$.

- If $\pi_k > \mu + \sigma$, we show that $\mathcal{J}_k((\pi^*(\mathcal{D}_v))_k) - \mathcal{J}_k(\pi_k) = \frac{1}{2}(\pi_k - (\pi^*(\mathcal{D}_v))_k) \geq \frac{\epsilon}{2}|\pi_k - (\pi^*(\mathcal{D}_v))_k|$.

- If $\pi_k \in [\mu, \mu + \sigma]$, we show that $\mathcal{J}_k((\pi^*(\mathcal{D}_v))_k) - \mathcal{J}_k(\pi_k) = \frac{\epsilon}{2}|(\pi^*(\mathcal{D}_v))_k - \pi_k|$.

Therefore, in any case, we have $\mathcal{J}_k((\pi^*(\mathcal{D}_v))_k) - \mathcal{J}_k(\pi_k) \geq \frac{\epsilon}{2}|\pi_k - (\pi^*(\mathcal{D}_v))_k|$. Similarly, we can show that this inequality holds for $v_k = 0$. Summing across coordinates $k = 1, \ldots, s$, we have the final claim. $\square$

**Step 4: Completing the proof.** We combine the previous steps to lower-bound the minimax risk by reducing the estimation problem to a multi-hypothesis testing problem using Fano's method in Theorem 3.3.

First, let $\mathcal{V} = \{v^{(1)}, \ldots, v^{(M)}\} \subset \{0, 1\}^s$ denote the fixed packing set constructed in Step 1. With notation abuse, we denote $J$ the random index drawn uniformly from $\{1, \ldots, M\}$, and define the random vector $V = v^{(J)} \in \mathcal{V}$. Conditioned on $V$, we draw a set of $N$ problem instances $S \sim \mathcal{D}_V^N$. This establishes the Markov chain $J \to V \to S$.

Using Fano's inequality, the expected estimation error from any learning algorithm $\pi$ is lower-bounded by:

$$\inf_\pi \sup_{v \in \mathcal{V}} \mathbb{E}[\|\pi(S) - \pi^*(\mathcal{D}_v)\|_1] \geq \delta\left(1 - \frac{I(J; S) + \log 2}{\log M}\right), \tag{2}$$

where $\delta = \Omega(\sigma s)$ is the geometric separation established in Lemma 5.7. From (1) and Lemma 5.8, we have

$$I(J; S) \leq \max_{i \neq j} \mathrm{KL}(\mathcal{D}_{v^{(i)}}^N \| \mathcal{D}_{v^{(j)}}^N) \leq 4Ns\epsilon^2.$$

Finally, we choose the perturbation parameter $\epsilon = \Theta\left(\frac{1}{\sqrt{N}}\right)$ to balance the information bound against the packing set capacity, ensuring that $I(J; S) \leq \frac{1}{2}\log M$. Substituting this choice into (2) guarantees that the testing error term $\left(1 - \frac{I(J;S) + \log 2}{\log M}\right)$ is bounded away from zero by a constant. Moreover, the choice of the perturbation scale $\epsilon$ also dictates the geometric separation $\delta$, we have $\delta = \Omega\left(\frac{s}{\sqrt{N}}\right)$. Combining this with Lemma 5.9, we have the final minimax lower bound of $\Omega\left(\frac{s}{\sqrt{N}}\right)$.

*Remark* 5.10 (Dependence on problem constants $B$ and $\pi_{\max}$). For clarity of the proof, the lower-bound construction in Theorem 5.6 intentionally uses normalized problem instances, effectively fixing $B = 1$ and $\pi_{\max} = 1$. However, we note that this construction can be easily extended to capture the explicit dependency on arbitrary parameter bounds. Specifically, by modifying the restricted problem class to use the scaled coupling matrix $A = B\mathbf{I}_s$, the constraint vector $b = \frac{B}{2}\mathbf{1}_s$, and the objective coefficients $c_k \in \{0, B\pi_{\max}\}$, the subsequent proof proceeds identically. This straightforward rescaling yields a minimax lower bound of $\Omega(B\pi_{\max}s/\sqrt{N})$, which confirms that the dependence on the problem constants $B$ and $\pi_{\max}$ on the upper bound in Theorem 5.5 is indeed tight.

*Remark* 5.11. Comparing the upper-bound of $\mathcal{O}(s^{1.5}/\sqrt{N})$ from Theorem 5.5 and the minimax lower-bound of $\Omega(s/\sqrt{N})$ from Theorem 5.6, we observe a gap of $\sqrt{s}$. While one might hypothesize that a more refined covering analysis that exploits the piecewise-linear structure of the Lagrangian dual function could tighten the ERM bound, the number of linear pieces $K$ can scale exponentially with $s$

(e.g., $K = 2^s$), rendering such structural arguments ineffective. This naturally raises the question: is the universal lower-bound loose, or does there exist an alternative learning algorithm that achieves the minimax optimal rate $\mathcal{O}(s/\sqrt{N})$? In the next section, we will answer this question constructively.

### 5.4. Minimax Optimality with Stochastic Gradient Ascent

We now demonstrate that the gap of $\sqrt{s}$ can be closed by shifting from ERM to an online-to-batch learning approach. Specifically, we show that Stochastic Gradient Ascent (SGA) with averaging (Algorithm 1) can achieve an expected excess risk that matches the minimax lower bound presented in Theorem 5.6.

---

**Algorithm 1** Stochastic Subgradient Ascent

> **Initialize:** $\pi_1 = \mathbf{0} \in \mathbb{R}^s$
> **for** $t = 1, \dots, N$ **do**
> > Receive a training MILP instance $P_t = (c_t, A_t, b_t, C_t, d_t) \sim \mathcal{D}$.
> > Solve the Lagrangian subproblem for $\pi_t$ to get $x_t^* \in \arg\min\{c_t^\top x + \pi_t^\top (b_t - A_t x) \ : \ x \in \mathbb{R}_+^m \times \{0, 1\}^p, \ C_t x \geq d_t\}$.
> > Compute the unbiased stochastic subgradient: $g_t = b_t - A_t x_t^*$.
> > Update and project onto the domain $\Pi = [0, \pi_{\max}]^s$: $\pi_{t+1} = \mathrm{Proj}_\Pi (\pi_t + \eta g_t)$.
> **end for**
> **Output:** The averaged multipliers $\bar{\pi}_N = \frac{1}{N} \sum_{t=1}^N \pi_t$.

---

Unlike ERM, which solves a static sample-average approximation, SGA processes the problem instances sequentially, updating the Lagrangian multipliers using unbiased stochastic (sub)gradients. By leveraging standard online learning regret bounds and the concavity and Lipschitzness of the expected utility function, we obtain the following guarantee.

**Theorem 5.12** (Minimax optimality of SGA). *If the SGA algorithm (Algorithm 1) is run for $N$ iterations with a constant step size $\eta = \frac{\pi_{\max}}{2B\sqrt{N}}$, the expected excess risk of the learned (averaged) Lagrangian multipliers $\bar{\pi}_N$ is upper-bounded by*

$$\mathbb{E}_{S \sim \mathcal{D}^N}[\mathcal{E}(\bar{\pi}_N)] \leq \frac{2B\pi_{\max}s}{\sqrt{N}} = \mathcal{O}\left(\frac{s}{\sqrt{N}}\right).$$

*Proof sketch.* The proof relies on standard online convex optimization (OCO) techniques. Since $g_t$ is an unbiased subgradient with $\|g_t\|_2 \leq 2B\sqrt{s}$, applying the projection update rule bounds the cumulative regret. Taking the expectation over the training problem instances and applying Jensen's inequality to the averaged multiplier $\bar{\pi}_N$ yields the final rate. See Appendix C.4 for the detailed proof. $\qquad\square$

Theorem 5.12 constructively shows that the $\Omega(s/\sqrt{N})$ lower bound established in Theorem 5.6 is fundamentally tight, completing the statistical picture for learning the Lagrangian multipliers via directly maximizing the utility function.

## 6. Learning to Warm-start Lagrangian Relaxation

In this section, we investigate the alternative paradigm of learning to warm-start, in which the objective shifts to minimizing the expected distance between the initial and optimal Lagrangian multipliers.

### 6.1. Problem Settings

We consider a setting in which a practitioner employs an iterative first-order solver, specifically subgradient ascent, to solve the Lagrangian dual problem for a new problem instance $P$. Our goal is to learn a deterministic initialization $\phi \in \Pi$ that minimizes the time to convergence. Classical results (Nesterov, 2018, Theorem 3.2.2) establish that for a non-smooth concave function with bounded subgradients, the number of iterations $T$ needed for subgradient ascent to reach an $\epsilon$-optimal solution, that is $\bar{\pi}$ such that $\ell(\bar{\pi}) - \ell^* \leq \epsilon$, is bounded by $T \propto L^2 \|\phi - \pi^*\|_2^2 / \epsilon^2$, where $L$ is the Lipschitz constant and $\phi$ is the initialization. Therefore, minimizing the squared Euclidean distance is a theoretically reasonable strategy for accelerating the solver.

To formulate a concrete learning objective, we must address the fact that the optimal Lagrangian multiplier for a MILP instance might *not be unique*. Let $\Pi^*(P) = \arg\max_{\pi \in \Pi} u(\pi, P)$ denote the set of optimal multipliers. To address the uniqueness concern, we can simply apply an arbitrary, but consistent tie-breaking rule when defining the optimal Lagrangian multipliers $\pi^*(P)$. For example, a natural choice is considering the *minimum norm solution*, that is,

$$\pi^*(P) \triangleq \arg\min_{\pi \in \Pi^*(P)} \|\pi\|_2^2.$$

Geometrically, this is equivalent to the projection onto the solution set, i.e., $\mathrm{Proj}_{\Pi^*(P)}(0_s)$. Note that, by the standard *Hilbert projection theorem* (Rudin, 1991, Theorem 12.3), because $\Pi^*(P)$ is closed and convex, this minimum norm projection is guaranteed to exist and be strictly unique. Motivated by this connection, we define the warm-start loss for an initialization $\phi$ on a problem instance $P$ as $\ell(\phi, P) \triangleq \|\phi - \pi^*(P)\|_2^2$.

**Objective of study.** Consistent with our framework in Section 4, we evaluate the performance of the learned initialization using expected excess risk. Formally, let $R(\phi) \triangleq \mathbb{E}_{P \sim \mathcal{D}}[\|\phi - \pi^*(P)\|_2^2]$ denote the population risk, we seek to bound the expected sub-optimality of the ERM

estimator $\hat{\phi}$:

$$\mathcal{E}(\hat{\phi}) \triangleq \mathbb{E}_{S \sim \mathcal{D}^N} \left[ R(\hat{\phi}(S)) - \min_{\phi \in \Pi} R(\phi) \right].$$

In contrast to the data-driven Lagrangian multiplier learning problem in Section 4, which involved maximizing a non-smooth concave function, this objective corresponds to a strongly convex minimization problem. As shown in the next section, this structure allows us to achieve a minimax optimality rate of $\Theta(s/N)$.

## 6.2. Minimax Optimality for Learning to Warm-start Lagrangian Relaxation

We now establish that the empirical mean estimator $\hat{\phi}(S)$ is minimax optimal for the learning-to-warm-start problem. We begin with the upper bound on the expected excess risk.

**Theorem 6.1** (Risk upper bound). *Let $S$ be a set of $N$ problem instances drawn i.i.d. from $\mathcal{D}$. The expected excess risk of the ERM estimator $\hat{\phi}(S)$ satisfies:*

$$\mathcal{E}(\hat{\phi}) = \mathcal{O}\left(\frac{s}{N}\right).$$

*Proof sketch.* The result follows by observing that for the squared Euclidean loss, the problem reduces to high-dimensional mean estimation, where $\hat{\phi}(S) = \frac{1}{N} \sum_{i=1}^{N} \pi^*(P_i)$ is simply the sample mean. Since the optimal multipliers lie on a bounded hypercube, applying Popoviciu's inequality (Niculescu & Popovici, 2006) to the trace of the estimator's covariance matrix gives the claim. Appendix D.1 presents the detailed derivation. □

Finally, we establish a minimax lower bound for the problem of learning to warm-start Lagrangian multipliers.

**Theorem 6.2** (Minimax lower-bound for learning to war-m-start). *For any learning algorithm $\phi$ taking input as a set $S$ of $N$ i.i.d. problem instances and output $\phi(S)$, the minimax expected risk is lower-bounded by*

$$\inf_{\phi} \sup_{\mathcal{D} \in \Delta(\mathcal{P})} \mathcal{E}(\phi) = \Omega\left(\frac{s}{N}\right).$$

*Proof sketch.* Our proof relies on a reduction to the fundamental problem of high-dimensional mean estimation. We sketch the main arguments:

- **Hard instance construction:** We adopt the construction Theorem 5.6 by restricting the problem instance to take the form $P = (c, \mathbf{I}_s, \frac{1}{2} \cdot \mathbf{1}_s, 0_{s \times t}, 0_t)$. From Lemma 5.7, we know that $\pi^*(P) = c$, if $c_k \geq 0$ for all $k$, for $P$ taking such restricted form.

- **Distribution family construction:** Again, we construct a family of distribution $\mathcal{D}_v$ parameterized by $v \in \{0, 1\}^s$ and a perturbation parameter $\epsilon$, using the set $\mathcal{V} \subset \{0, 1\}^s$. Again, from Lemma C.1, $|\mathcal{V}| \geq 2^{s/8}$ and $d_H(v^{(i)}, v^{(j)}) \geq \frac{s}{8}$. However, a key distinction is that the *warm-start objective is squared loss*, allowing us to show that $\|\phi^*(\mathcal{D}_{v^{(j)}}) - \phi^*(\mathcal{D}_{v^{(i)}})\|_2^2 \geq \frac{s\epsilon^2}{8}$.

- **Minimax lower-bound**: Finally, to satisfy the mutual information condition in Fano's method, we choose $\epsilon \asymp \frac{1}{\sqrt{N}}$. Substituting Theorem 3.3, we have the final claim.

We note that the construction above naturally enforces that the optimal Lagrangian multiplier given a problem instance is strictly unique. The detailed proof is presented in Appendix D. □

*Remark* 6.3 (Statistical advantage of warm-starting). Comparing the minimax lower bound of $\Omega(s/N)$ derived from Theorem 6.2 with the $\Omega(s/\sqrt{N})$ bound for direct multiplier learning from Theorem 5.6, we observe a fundamental gap in the sample complexity: the learning to warm-start achieves a fast rate and minimax optimality. This improvement arises naturally because we effectively replace a non-smooth maximization problem with a strongly convex mean estimation problem. This explains why warm-start strategies are often more sample-efficient and more robust than direct Lagrangian dual maximization.

# 7. Conclusion and Future Works

We established the first rigorous statistical foundation for data-driven Lagrangian relaxation in MILP. Our analysis characterizes the sample complexity of learning multipliers, deriving an upper bound of $\mathcal{O}(s^{1.5}/\sqrt{N})$ and a minimax lower bound of $\Omega(s/\sqrt{N})$. We constructively close this gap by demonstrating that Stochastic Gradient Ascent (SGA) with averaging strictly achieves the minimax optimal rate of $\Theta(s/\sqrt{N})$. Crucially, we demonstrated that shifting the objective from direct dual maximization to learning to warm-start fundamentally alters the problem geometry—from non-smooth concave maximization to strongly convex mean estimation—thereby unlocking a fast, minimax-optimal rate of $\Theta(s/N)$. Consequently, these results provide theoretical justification for the empirical success of learning-based acceleration strategies in discrete optimization.

Our work opens several interesting directions for future work. While our current framework relies on problem instances drawn from a stationary problem distribution, extending our analysis to handle shifts in the problem distribution remains an important open question. Furthermore, beyond Lagrangian relaxation, analyzing the sample complexity of other classical decomposition techniques, such as Dantzig-Wolfe or Bender decomposition, is an exciting theoretical question for the learning-to-optimize line of work.

## Acknowledgments

Research reported in this paper was partially supported through the French ANR through the MIAI Cluster (reference ANR-23-IACL-0006). Viet Anh Nguyen gratefully acknowledges the support from the CUHK's Improvement on Competitiveness in Hiring New Faculties Funding Scheme, UGC ECS Grant 24210924, and UGC GRF Grant 14208625.

## Impact Statement

This paper presents work whose goal is to advance the field of AI and Machine Learning. There are many potential societal consequences of our work, none of which we feel must be specifically highlighted here.

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

## A. Vehicle Routing Problem and Its Decomposition

Consider the vehicle routing problem, where

- $V$ denotes the set of nodes, each of which represents a customer or the depot,

- $K$ is the set of vehicles,

- $c_{ij}$ is the cost (e.g., distance) to travel from node $i$ to node $j$, where $i, j \in V$,

- $d_i$ denotes the demand of customer $i$, and

- $Q$ denotes the capacity of each vehicle (assume that all vehicles $k \in K$ have the same capacity).

The decision variables is $x \in \{0, 1\}^{|V| \times |V| \times |K|}$, where $x_{ijk}$ equals to 1 if the vehicle $k \in K$ travels directly from node $i$ to node $j$, and 0 otherwise. Then, the vehicle routing problem is formulated as follows

$$\min \sum_{i \in V, j \in V, k \in K} c_{ij} x_{ijk}$$

$$\text{s.t. } x \in \{0, 1\}^{|V| \times |V| \times |K|} \tag{3a}$$

$$\sum_{k \in K} \sum_{j \in V} x_{jik} = 1 \qquad \forall i \in V \setminus \{0\} \tag{3b}$$

$$\sum_{j \in V \setminus \{0\}} x_{0jk} = 1 \qquad \forall k \in K \tag{3c}$$

$$\sum_{i \in V \setminus \{0\}} x_{i0k} = 1 \qquad \forall k \in K \tag{3d}$$

$$\sum_{j \in V} x_{jik} - \sum_{j \in V} x_{ijk} = 0 \qquad \forall i \in V \setminus \{0\}, \forall k \in K \tag{3e}$$

$$\sum_{i \in V \setminus \{0\}} d_i \left( \sum_{j \in V} x_{jik} \right) \leq Q \qquad \forall k \in K. \tag{3f}$$

Here, the constraint (3b) is the linking constraint, ensuring that each customer is visited exactly once by a single vehicle. The constraints (3c) and (3d) ensure that every vehicle must depart from the depot and must return to the depot. The constraint (3e) means that if a vehicle arrives at a customer, it must also depart from the same customer. And the final constraint (3f) implies that the total customer demand on any single vehicle's route must not exceed its capacity. We have omitted the subtour elimination constraints for each vehicle to simplify exposition.

Note that the constraint (3b) is the crucial link that imposes the relation between vehicles. If we dualize such a constraint with Lagrangian multipliers, we can decompose the original complicated problems into many easier-to-solve sub-problems, each corresponding to a vehicle $k \in K$. Concretely, let $f(\boldsymbol{\pi})$, where $\boldsymbol{\pi} \in \mathbb{R}^{|V|-1}$, be the objective function of the Lagrangian relaxation problem corresponding to the multiplier $\pi$, we have

$$f(\pi) = \sum_{i \in V, j \in V, k \in K} c_{ij} x_{ijk} + \sum_{i \in V \setminus \{0\}} \pi_i \left( \sum_{k \in K} \sum_{j \in V} x_{jik} - 1 \right)$$

$$= \sum_{k \in K} \left( \sum_{i \in V, j \in V} c_{ij} x_{ijk} + \sum_{i \in V \setminus \{0\}, j \in V} \pi_i x_{jik} \right) - \sum_{i \in V \setminus \{0\}} \pi_i.$$

Moreover, note that the other constraints (constraints (3c), (3d), (3e), (3f)) are defined for each vehicle $k \in K$. Therefore,

given the multiplier $\pi$, we can decompose the original problems into many sub-problems $\boldsymbol{P}_k$, for $k \in K$, as follow

$$
\begin{aligned}
\min \quad & \sum_{i \in V, j \in V} c_{ij} x_{ijk} + \sum_{i \in V \setminus \{0\}, j \in V} \pi_i x_{jik} \\
\text{s.t.} \quad & x_k \in \{0, 1\}^{|V| \times |V|} \\
& \sum_{j \in V \setminus \{0\}} x_{0jk} = 1, \quad \sum_{i \in V \setminus \{0\}} x_{i0k} = 1 \\
& \sum_{j \in V} x_{jik} - \sum_{j \in V} x_{ijk} = 0 \quad \forall i \in V \setminus \{0\} \\
& \sum_{i \in V \setminus \{0\}} d_i \left( \sum_{j \in V} x_{jik} \right) \leq Q.
\end{aligned}
\tag{$\boldsymbol{P}_k$}
$$

## B. Additional Backgrounds

First, we recall Talagrand's contraction inequality, which is crucial for analyzing the Rademacher complexity of a class of composite functions involving Lipschitz functions.

**Lemma B.1** (Talagrand's contraction inequality, Ledoux & Talagrand (1991)). *Let $\mathcal{U}$ be a real-valued function class that take inputs from domain $\mathcal{P}$, and let $S = \{P_1, \ldots, P_N\} \subset \mathcal{P}$ be a set of $N$ samples. Let $\phi_1, \ldots, \phi_N$ be a set of L-Lipschitz functions (i.e., $|\phi_i(a) - \phi_i(b)| \leq L |a - b|$ for all $a, b \in \mathbb{R}$). Then we have*

$$
\mathbb{E}_\sigma \left[ \sup_{u \in \mathcal{U}} \frac{1}{N} \sum_{i=1}^N \sigma_i \phi_i(u(P_i)) \right] \leq L \cdot \mathbb{E}_\sigma \left[ \sup_{u \in \mathcal{U}} \frac{1}{N} \sum_{i=1}^N \sigma_i u(P_i) \right],
$$

*where $\sigma = (\sigma_1, \ldots, \sigma_N)$ are i.i.d. Rademacher random variables.*

We recall the vector Bernstein's inequality, a concentration inequality with controlled variance.

**Lemma B.2** (Vector Bernstein's inequality, Bernstein (1924)). *Let $X_1, \ldots, X_N$ be independent zero-mean random vectors in $\mathbb{R}^d$. Suppose that $\|X_i\|_2 \leq M$ almost surely for all $i \in \{1, \ldots, N\}$. Let $\sigma^2 = \sum_{i=1}^N \mathbb{E}[\|X_i\|_2^2]$ be the total variance. Then for any $t > 0$,*

$$
\mathbb{P}\left( \left\| \sum_{i=1}^N X_i \right\|_2 > t \right) \leq (d+1) \exp\left( \frac{-t^2}{2(\sigma^2 + Mt/3)} \right).
$$

We then recall Popoviciu's inequality, a preliminary result that provides an upper bound for the variance of random variables with bounded support.

**Lemma B.3** (Popoviciu's inequality, Popoviciu (1965)). *Let $X$ be a random variable such that it is bounded by a minimum value $m$ and a maximum value $M$. Then $\mathrm{Var}(X) \leq \frac{(M-m)^2}{4}$.*

To establish an upper bound on the generalization guarantee of data-driven learning Lagrangian multipliers, we use several tools to control the covering number and Rademacher complexity. For completeness, we provide their formal definitions and statements as follows.

**Definition B.4** (Empirical $L^2(\mathcal{P}^N)$-metric, Wainwright (2019)). Let $\mathcal{U}$ be a function class of which each function takes input in $\mathcal{P}$. Let $S = \{P_1, \ldots, P_N\} \subset \mathcal{P}$ be a set of $N$ problem instances. We define the *empirical $L^2(\mathcal{P}^N)$-metric* on $\mathcal{U}$ as

$$
\|u - u'\|_{2,N} \coloneqq \sqrt{\frac{1}{N} \sum_{i=1}^N (u(P_i) - u'(P_i))^2}.
$$

**Definition B.5** (Covering set and covering number, Wainwright (2019)). Let $(X, \|\cdot\|)$ be a metric space, and let $\Pi \subset X$ be a subset of $X$. We say that $\Pi' \subset \Pi$ is a $\delta$-covering set of $\Pi$ if for any $\pi \in \Pi$, there exists $\pi' \in \Pi'$ such that $\|\pi - \pi'\| \leq \delta$. A minimum $\delta$-covering set of $\Pi$ is the $\delta$-covering set of $\Pi$ that has the fewest elements. The $\delta$-covering number of $\Pi$, denote $\mathcal{N}(\delta, \Pi, \|\cdot\|)$ is the number of elements of a minimum $\delta$-covering set of $\Pi$.

**Lemma B.6** (Dudley's entropic integral, Wainwright (2019)). *Let $D = \sup_u \|u\|_{2,N}$, we have*

$$\hat{\mathscr{R}}_N(\mathcal{U}) \leq \inf_{\delta_0 > 0} \delta_0 + \int_{\delta_0}^{D} \sqrt{\frac{\log \mathcal{N}(\delta, \mathcal{U}, \|\cdot\|_{2,N})}{N}} \, d\delta.$$

# C. Proofs for Section 5

In this section, we present a detailed proof of the minimax lower bound for the data-driven learning Lagrangian relaxation problem.

## C.1. Additional Backgrounds

First, we recall a classical result for constructing the packing set for an $s$-dimensional hypercube.

**Lemma C.1** (Varshamov-Gilbert, Lemma 2.9, (Tsybakov, 2008)). *Let $s \geq 8$. There exists a subset $\mathcal{M} = \{v^{(1)}, \ldots, v^{(M)}\} \subset \{0,1\}^s$ such that:*

- $M \geq 2^{s/8}$.

- $v^{(1)} = (0, \ldots, 0) \in \{0,1\}^s$.

- *For any distinct pair $v^{(i)}, v^{(j)} \in \mathcal{M}$, the Hamming distance $d_H(v^{(i)}, v^{(j)}) \geq \frac{s}{8}$,*

- *Consequently, the $\ell_1$ distance between any distinct pair satisfies $\|v^{(i)} - v^{(j)}\|_1 \geq \frac{s}{8}$.*

## C.2. Proof of the Geometric Properties

*Proof of Proposition 5.1.* We first prove the concavity property. Recall that for $P = (c, A, b, C, d)$, the optimal value of the corresponding Lagrangian relaxation $u(\pi, P)$ is defined as

$$u(\pi, P) = \min_{x \in \mathcal{X}} L(\pi, x),$$

where $L(\pi, x) = c^\top x + \pi^\top (b - Ax)$ is the Lagrangian function and $\mathcal{X} = \{x \in \mathbb{R}_+^m \times \{0,1\}^p \mid Cx \geq d\}$. Note that $L(\pi, x)$ is an affine function of $\pi$ and therefore also concave in $\pi$. This means that $u(\pi, P)$ is the point-wise minimum of concave functions of $\pi$, hence it is also concave.

Next, we prove that $g(\pi, P) = b - Ax^*(\pi, P)$ is a valid subgradient of $u(\pi, P)$. First, note that $u(\cdot, P)$ is a concave function of $\pi$. Therefore, a vector $g$ is a valid subgradient of $u$ at $\pi$ if for any multiplier $\pi' \in \mathbb{R}_+^s$, we have

$$u(\pi', P) \leq u(\pi, P) + g^\top (\pi' - \pi).$$

Now let $x^*(\pi, P)$ be the optimal solution that achieves the minimum for $u(\pi, P)$. By definition, we have $u(\pi, P) = c^\top x^*(\pi, P) + \pi^\top (b - Ax^*(\pi, P))$. Now, consider any $\pi' \in \mathbb{R}_+^s$, by the definition of $u(\pi', P)$, we have

$$\begin{aligned}
u(\pi', P) &= \min_{x \in \mathcal{X}} c^\top x + (\pi')^\top (b - Ax) \\
&\leq c^\top x^*(\pi, P) + (\pi')^\top (b - Ax^*(\pi, P)) \\
&= \left[ c^\top x^*(\pi, P) + \pi^\top (b - Ax^*(\pi, P)) \right] + (\pi' - \pi)^\top (b - Ax^*(x, P)) \\
&= u(\pi, P) + g(\pi, P)^\top (\pi' - \pi).
\end{aligned}$$

This means that given a problem instance $P$, $g(\pi, P) = b - Ax^*(\pi, P)$ is a valid sub-gradient of $u(\pi, P)$ for any $\pi \in \mathbb{R}_+^s$. Finally, from Assumption 4.1, we have

$$\|g(\pi, P)\|_2 = \|b - Ax^*(\pi, P)\|_2 \leq \|b\|_2 + \|Ax^*(\pi, P)\|_2 \leq 2B\sqrt{s}.$$

The proof is complete. $\qquad\square$

## C.3. Minimax Lower-bound for Data-driven Learning Lagrangian Relaxation

We now present the detailed proof of the minimax lower bound for the data-driven learning Lagrangian relaxation problem.

*Proof of Theorem 5.6.* We break down the proof into the following steps.

**Step 1 - Restricting the problem space:** First, throughout this construction, we consider the problem instance $P$ which is strictly in the form $P = (c, \mathbf{I}_s, \frac{1}{2} \cdot \mathbf{1}_s, 0_{t \times s}, 0_t)$ representing the ILP problem

$$\min_{x \in \{0,1\}^s} c^\top x \quad \text{s.t.} \quad x_k \geq \frac{1}{2} \quad \forall k = 1, \ldots, s.$$

Here, $(C, d) = (0_{t \times s}, 0_t)$ means all constraints are hard constraints, i.e., they will all be dualized in the Lagrangian relaxation problem. Besides, the problem instance $P$ is solely determined by the objective vector $c$, meaning that we can equate the problem distribution $\mathcal{D}$ for $P$ as a distribution for $c$. Besides, for any $\pi \in \mathbb{R}_+^s$, we have

$$u(\pi, P) = \min_{x \in \{0,1\}^s} c^\top x + \pi^\top (\frac{1}{2} \cdot \mathbf{1}_s - x) = \frac{1}{2} \pi^\top \mathbf{1}_s + \min_{x \in \{0,1\}^s} (c - \pi)^\top x$$

$$= \frac{1}{2} \pi^\top \mathbf{1}_s + \sum_{k=1}^s \min(0, c_k - \pi_k) = \sum_{k=1}^s \min \left( \frac{1}{2} \pi_k, c_k - \frac{\pi_k}{2} \right).$$

**Step 2 - Defining problem distribution:** Let $\mu > 0$ be a based constant and $\sigma > 0$ be a scale parameter, where $\mu + \sigma < \pi_{\max}$. For a fixed vector $v \in \{0, 1\}^s$, we define the problem distribution $\mathcal{D}_v$ over $c$ as follow:

- Support: $c_k$ takes value in $\{\mu, \mu + \sigma\}$ for any $k = 1, \ldots, s$,

- Probabilities:

    - If $v_k = 1$, then $\mathbb{P}(c_k = \mu + \sigma) = \frac{1+\epsilon}{2} = p_k$, and $\mathbb{P}(c_k = \mu) = \frac{1-\epsilon}{2} = 1 - p_k$,
    - If $v_k = 0$, then $\mathbb{P}(c_k = \mu + \sigma) = \frac{1-\epsilon}{2} = p_k$, and $\mathbb{P}(c_k = \mu) = \frac{1+\epsilon}{2} = 1 - p_k$.

Here, $\epsilon > 0$ is a tunable parameter. Note that with the problem distribution $\mathcal{D}_v$, the optimal Lagrangian relaxation $\pi^*(\mathcal{D}_v)$ is

$$\pi^*(\mathcal{D}_v) = \arg \max_{\pi \in \Pi} \mathbb{E}_{c \sim \mathcal{D}_v} \left[ \sum_{k=1}^s \min \left( \frac{1}{2} \pi_k, c_k - \frac{\pi_k}{2} \right) \right].$$

Note that this optimization problem can be decomposed into $s$ sub-problems, i.e., $(\pi^*(\mathcal{D}_v))_k$ can be defined as

$$(\pi^*(\mathcal{D}_v))_k = \arg \max_{\pi_k} \left\{ \mathcal{J}_k(\pi_k) \triangleq \mathbb{E}_{c_k \sim \mathcal{D}_{v_k}} \left[ \min \left( \frac{1}{2} \pi_k, c_k - \frac{\pi_k}{2} \right) \right] \right\}$$

We then have the following cases

- Case 1: $v_k = 1$, then $p_k > \frac{1}{2}$. We have

$$\mathcal{J}_k(\pi_k) = p_k \cdot \min \left( \frac{1}{2} \pi_k, \mu + \sigma - \frac{\pi_k}{2} \right) + (1 - p_k) \cdot \min \left( \frac{1}{2} \pi_k, \mu - \frac{\pi_k}{2} \right),$$

    and we consider the following cases:

    - Case 1.1: $\pi_k < \mu$, then $\mathcal{J}_k(\pi_k) = \frac{\pi_k}{2}$ and $\frac{\partial \mathcal{J}_k}{\partial \pi_k} = \frac{1}{2}$. This means $\mathcal{J}_k$ is increasing over $[0, \mu]$.
    - Case 1.2: $\pi_k > \mu + \sigma$, then $\mathcal{J}_k(\pi_k) = p_k \left( \mu + \sigma - \frac{\pi_k}{2} \right) + (1 - p_k) \left( \mu - \frac{\pi_k}{2} \right)$, and $\frac{\partial \mathcal{J}_k}{\partial \pi_k} = -\frac{1}{2}$. This means $\mathcal{J}_k$ is decreasing over $[\mu + \sigma, \infty)$.
    - Case 1.3: $\pi_k \in [\mu, \mu + \sigma]$, then $\mathcal{J}_k(\pi_k) = p_k \cdot \frac{\pi_k}{2} + (1 - p_k)(\mu - \frac{\pi_k}{2})$ and $\frac{\partial \mathcal{J}_k}{\partial \pi_k} = p_k - \frac{1}{2}$. Since $p_k > \frac{1}{2}$, this means $\mathcal{J}_k$ is increasing over $[\mu, \mu + \sigma]$.

Therefore, $\mathcal{J}_k$ attains its maxima at $\pi_k = \mu + \sigma$, i.e., $(\pi^*(\mathcal{D}_v))_k = \mu + \sigma$.

- Case 2: $v_k = 0$, then $p_k < \frac{1}{2}$. Using the same idea as the previous case, we have $(\pi^*(\mathcal{D}_v))_k = \mu$.

Then, we conclude that

$$\pi^*(\mathcal{D}_v) = \arg\max_{\pi \in \Pi} \mathbb{E}_{c \sim \mathcal{D}_v} \sum_{k=1}^{s} \min\left(\frac{1}{2}\pi_k, c_k - \frac{\pi_k}{2}\right) = \mu\mathbf{1}_s + \sigma v.$$

**Step 3 - Defining a family of hard problem distributions**: Let $\{0, 1\}^s$ be an $s$-dimensional hypercube. From Lemma C.1, there exists a subset $\mathcal{V} = \{v^{(1)}, \ldots, v^{(M)}\} \subset \{0, 1\}^s$ such that $M \geq 2^{s/8}$ and $\|v^{(i)} - v^{(j)}\|_1 \geq \frac{s}{8}$. This means that for any distinct $v^{(i)}, v^{(j)} \in \mathcal{V}$, we have

$$\|\pi^*(\mathcal{D}_{v^{(i)}}) - \pi^*(\mathcal{D}_{v^{(j)}})\|_1 = \|\mu\mathbf{1}_s + \sigma v^{(i)} - \mu\mathbf{1}_s - \sigma v^{(j)}\|_1 = \sigma\|v^{(i)} - v^{(j)}\|_1 \geq \frac{\sigma s}{8}.$$

This means that the set $\{\pi^*(\mathcal{D}_{v^{(1)}}), \ldots, \pi^*(\mathcal{D}_{v^{(M)}})\}$ is a $\frac{\sigma s}{16}$-packing set of $\Pi$. Besides, the set $\mathcal{V}$ defines a family of hard problem distribution $\{\mathcal{D}_{v^{(1)}}, \ldots, \mathcal{D}_{v^{(M)}}\}$, and the minimax risk is lower-bounded by

$$\inf_{\pi} \sup_{\mathcal{D} \in \Delta(\mathcal{P})} \mathbb{E}_{S \sim \mathcal{D}^N}[U(\pi^*(\mathcal{D}), \mathcal{D}) - U(\pi(S), \mathcal{D})] \geq \inf_{\pi} \sup_{v \in \mathcal{V}} \mathbb{E}_{S \sim \mathcal{D}_v^N}[U(\pi^*(\mathcal{D}_v), \mathcal{D}_v) - U(\pi(S), \mathcal{D}_v)].$$

**Step 4 - Linking the risk with the distance in the parameter space**: We now show that for any $\pi_k$, we have

$$\mathcal{J}_k((\pi^*(\mathcal{D}_v))_k) - \mathcal{J}_k(\pi_k) \geq \frac{\epsilon}{2}\left|(\pi^*(\mathcal{D}_v))_k - \pi_k\right|.$$

If $v_k = 1$, we consider the following case:

- If $\pi_k < \mu$: We have

$$\begin{aligned}
\mathcal{J}_k((\pi^*(\mathcal{D}_v))_k) - \mathcal{J}_k(\pi_k) =& \mathcal{J}_k((\pi^*(\mathcal{D}_v))_k) - \mathcal{J}_k(\mu) + \mathcal{J}_k(\mu) - \mathcal{J}_k(\pi_k) \\
=& \frac{\epsilon}{2}((\pi^*(\mathcal{D}_v))_k - \mu) + \frac{1}{2}(\mu - \pi_k) \\
\geq& \frac{\epsilon}{2}((\pi^*(\mathcal{D}_v))_k - \mu) + \frac{\epsilon}{2}(\mu - \pi_k) = \frac{\epsilon}{2}((\pi^*(\mathcal{D}_v))_k - \pi_k).
\end{aligned}$$

- If $\pi_k > \mu + \sigma$: We have

$$\begin{aligned}
\mathcal{J}_k((\pi^*(\mathcal{D}_v))_k) - \mathcal{J}_k(\pi_k) =& p_k \cdot \frac{\mu + \sigma}{2} + (1 - p_k)\left(\mu - \frac{\mu + \sigma}{2}\right) - p_k\left(\mu + \sigma - \frac{\pi_k}{2}\right) - (1 - p_k)\left(\mu - \frac{\pi_k}{2}\right) \\
=& p_k\left(-\frac{\mu + \sigma}{2} + \frac{\pi_k}{2}\right) + (1 - p_k)\left(\frac{\pi_k}{2} - \frac{\mu + \sigma}{2}\right) \\
=& \frac{\pi_k}{2} - \frac{\mu + \sigma}{2} \\
=& \frac{1}{2}(\pi_k - (\pi^*(\mathcal{D}_v))_k) \qquad \text{(as } (\pi^*(\mathcal{D}_v))_k = \mu + \sigma) \\
\geq& \frac{\epsilon}{2}(\pi_k - (\pi^*(\mathcal{D}_v))_k) \qquad \text{(as } \epsilon \text{ is small).}
\end{aligned}$$

- If $\pi_k \in [\mu, \mu + \sigma]$: From the above, in such a case, we have $\frac{\partial \mathcal{J}_k}{\partial \pi_k} = p_k - \frac{1}{2} = \frac{\epsilon}{2}$ for any $\pi_k \in [\mu, \sigma]$. And since $\mathcal{J}_k$ is linear over $[\mu, \mu + \sigma]$ and $(\pi^*(\mathcal{D}_v))_k = \mu + \sigma$, we claim that $\mathcal{J}_k((\pi^*(\mathcal{D}_v))_k) - \mathcal{J}(\pi_k) = \frac{\epsilon}{2}|(\pi^*(\mathcal{D}_v))_k - \pi_k|$.

Therefore, if $v_k = 1$, we can claim that $\mathcal{J}_k((\pi^*(\mathcal{D}_v))_k) - \mathcal{J}_k(\pi_k) \geq \frac{\epsilon}{2}|(\pi^*(\mathcal{D}_v))_k - \pi_k|$. Using an analogous argument, we can claim that for $v_k = 0$, we also have $\mathcal{J}_k((\pi^*(\mathcal{D}_v))_k) - \mathcal{J}_k(\pi_k) \geq \frac{\epsilon}{2}|(\pi^*(\mathcal{D}_v))_k - \pi_k|$. Take the sum across the

index $k = 1, \ldots, s$, and note that $P$ is determined by only the objective vector $c$ (hence we can write $c \sim \mathcal{D}_v$ instead of $P \sim \mathcal{D}_v$), we have

$$U(\pi^*(\mathcal{D}_v), \mathcal{D}_v) - U(\hat{\pi}, \mathcal{D}_v) = \mathbb{E}_{c \sim \mathcal{D}_v}\left[u(\pi^*(\mathcal{D}_v)) - u(\pi, P)\right]$$

$$= \sum_{k=1}^{s} \mathbb{E}_{c \sim \mathcal{D}_v}[\mathcal{J}_k((\pi^*(\mathcal{D}_v))_k) - \mathcal{J}_k(\pi_k)] \geq \frac{\epsilon}{2}\|(\pi^*(\mathcal{D}_v) - \pi\|_1.$$

Combining with the claim from Step 3, we have

$$\inf_{\pi} \sup_{\mathcal{D} \in \Delta(\mathcal{P})} \mathbb{E}_{S \sim \mathcal{D}^N}[U(\pi^*(\mathcal{D}), \mathcal{D}) - U(\pi(S), \mathcal{D})] \geq \frac{\epsilon}{2} \inf_{\pi} \sup_{v \in \mathcal{V}} \mathbb{E}_{S \sim \mathcal{D}_v^N}\|\pi^*(\mathcal{D}_v) - \pi(S)\|_1.$$

**Step 5: Applying Fano's method (Theorem 3.3):** We now use Fano's method to give a lower bound for the LHS above. We have

- From Lemma C.1, the log cardinality of the packing set $\mathcal{V}$ is lower-bounded by $\log M \geq \frac{s}{8} \log 2$.

- *Mutual information upper-bound*: In our construction, the coordinate of objective vector $c$ are independent, the KL divergence sums over $s$ dimensions. For any distinct $v^{(i)}, v^{(j)} \in \mathcal{V}$, the KL divergence between $\mathcal{D}_{v^{(i)}}$ and $\mathcal{D}_{v^{(j)}}$ is upper-bounded by the sum of $s$ Bernoulli KL divergences. For small $\epsilon \in (0, 1/2)$, using standard $\chi^2$-upper bound for KL divergence, we have $\mathrm{KL}(\mathrm{Ber}(\frac{1+\epsilon}{2})\|\mathrm{Ber}(\frac{1-\epsilon}{2})) \leq \chi^2(P\|Q) = \frac{4\epsilon^2}{1-\epsilon^2} = \mathcal{O}(\epsilon^2)$ using $\chi^2$-upper bound. Therefore, we have $I(J; S) \leq N \max_{i,j} \mathrm{KL}(\mathcal{D}_{v^{(i)}}\|\mathcal{D}_{v^{(j)}}) \leq Ns(4\epsilon^2)$.

To make Fano's inequality valid, we need to choose $\epsilon$ such that

$$\frac{4Ns\epsilon^2 + \log 2}{\frac{s}{8}\log 2} \leq \frac{1}{2} \Leftrightarrow 4Ns\epsilon^2 \leq (\frac{s}{16} - 1)\log 2.$$

We simply choose $\epsilon \asymp \frac{1}{\sqrt{N}}$. Combining with the claim from Steps 3, 4, and applying Theorem 3.3, we have

$$\inf_{\pi} \sup_{\mathcal{D} \in \Delta(\mathcal{P})} \mathbb{E}_{S \sim \mathcal{D}^N}[U(\pi^*(\mathcal{D}), \mathcal{D}) - U(\pi(S), \mathcal{D})] \geq \frac{\epsilon}{2} \inf_{\pi} \sup_{v \in \mathcal{V}} \mathbb{E}_{S \sim \mathcal{D}_v^N}\|\pi^*(\mathcal{D}_v) - \pi(S)\|_1 = \Omega\left(\frac{1}{\sqrt{N}} \cdot \sigma s\right),$$

which concludes the proof. $\square$

## C.4. Minimax Optimality via Stochastic Gradient Ascent

In this section, we will present the formal proof for Theorem 5.12.

**Theorem 5.12 (restated).** *If the SGA algorithm (Algorithm 1) is run for $N$ iterations with a constant step size $\eta = \frac{\pi_{\max}}{2B\sqrt{N}}$, the expected excess risk of the learned (averaged) Lagrangian multipliers $\bar{\pi}_N$ is upper-bounded by*

$$\mathbb{E}_{S \sim \mathcal{D}^N}[\mathcal{E}(\bar{\pi}_N)] \leq \frac{2B\pi_{\max}s}{\sqrt{N}} = \mathcal{O}\left(\frac{s}{\sqrt{N}}\right).$$

*Proof of Theorem 5.12.* Let $U(\pi) = \mathbb{E}_{P \sim \mathcal{D}}[u(\pi, P)]$ denote the expected utility function corresponding to Lagrangian variables $\pi$. Since $u(\cdot, P)$ is concave for any problem instance $P$, $U(\pi)$ is also concave. Besides, recall that $\pi^*(\mathcal{D}) \in \arg\max_{\pi \in \Pi} U(\pi)$ is the optimal Lagrangian variable corresponding to the problem distribution $\mathcal{D}$.

From the property of the projection operator onto a convex set $\Pi$, for any $t \in \{1, \ldots, N\}$, we have

$$\|\pi_{t+1} - \pi^*(\mathcal{D})\|_2^2 \leq \|\pi_t + \eta g_t - \pi^*(\mathcal{D})\|_2^2 = \|\pi_t - \pi^*(\mathcal{D})\|_2^2 + 2\eta g_t^\top(\pi_t - \pi^*(\mathcal{D})) + \eta^2\|g_t\|_2^2$$

$$\Rightarrow \quad g_t^\top(\pi^*(\mathcal{D}) - \pi_t) \leq \frac{\|\pi_t - \pi^*(\mathcal{D})\|_2^2 - \|\pi_{t+1} - \pi^*(\mathcal{D})\|_2^2}{2\eta} + \frac{\eta}{2}\|g_t\|_2^2. \tag{4}$$

Note that $u(\cdot, P_t)$ is a concave function, therefore $u(\pi^*(\mathcal{D}), P_t) - u(\pi_t, P_t) \leq g_t^\top(\pi^*(\mathcal{D}) - \pi_t)$. Note that $\pi_t$ is constructed using only the historical instances $\{P_1, \ldots, P_{t-1}\}$, hence is independent with $P_t$. Therefore, if we take the expectation over

all problem instances $P_1, \ldots, P_N \sim \mathcal{D}$, then $\mathbb{E}_{P_1, \ldots, P_N}[u(\pi_t, P)] = \mathbb{E}_{P_1, \ldots, P_N}[U(\pi_t)]$, and $\mathbb{E}_{P_1, \ldots, P_N}[u(\pi^*(\mathcal{D}), P)] = U(\pi^*)$ since $\pi^*$ is deterministic. Combining the above with Equation 4, we have

$$\mathbb{E}_{P_1, \ldots, P_N}[U(\pi^*) - U(\pi_t)] \leq \mathbb{E}_{P_1, \ldots, P_N}\left[\frac{\|\pi_t - \pi^*(\mathcal{D})\|_2^2 - \|\pi_{t+1} - \pi^*(\mathcal{D})\|_2^2}{2\eta} + \frac{\eta}{2}\|g_t\|_2^2\right].$$

Taking the sum over $N$ time steps, we have

$$\mathbb{E}_{P_1, \ldots, P_N}\left[NU(\pi^*) - \sum_{t=1}^N U(\pi_t)\right] \leq \mathbb{E}_{P_1, \ldots, P_N}\left[\frac{\|\pi_1 - \pi^*(\mathcal{D})\|_2^2 - \|\pi_{N+1} - \pi^*(\mathcal{D})\|_2^2}{2\eta} + \frac{\eta}{2}\sum_{t=1}^N \|g_t\|_2^2\right]$$

$$\leq \frac{\|\pi_1 - \pi^*(\mathcal{D})\|_2^2}{2\eta} + \mathbb{E}_{P_1, \ldots, P_N}\left[\sum_{t=1}^N \|g_t\|_2^2\right],$$

where we use the fact that $\pi_1$ is fixed, so $\mathbb{E}_{P_1, \ldots, P_N}\|\pi_1 - \pi^*(\mathcal{D})\|_2^2 = \|\pi_1 - \pi^*(\mathcal{D})\|_2^2$. Now note that $\|g_t\|_2 \leq 2B\sqrt{s}$, and $\|\pi_1 - \pi^*(\mathcal{D})\|_2^2 \leq s\pi_{\max}^2$, we have

$$\mathbb{E}_{P_1, \ldots, P_N}\left[U(\pi^*) - \sum_{t=1}^N U(\pi_t)\right] \leq \frac{s\pi_{\max}^2}{2\eta} + 2\eta N B^2 s.$$

Now, since $U(\cdot)$ is a concave function, using Jensen's inequality, we have $\frac{1}{N}\sum_{t=1}^N U(\pi_t) \leq U\left(\frac{1}{N}\sum_{t=1}^N \pi_t\right) = U(\bar{\pi}_N)$, and therefore

$$N\mathbb{E}[\mathcal{E}(\bar{\pi}_N)] = N\mathbb{E}_{P_1, \ldots, P_N}[U(\pi^*(\mathcal{D})) - U(\bar{\pi}_N)] \leq \frac{s\pi_{\max}^2}{2\eta} + 2\eta N B^2 s.$$

Choosing $\eta = \frac{\pi_{\max}}{2B\sqrt{N}}$, we have the final conclusion. □

# D. Proofs for Section 6

## D.1. Upper Bound

We now present the detailed proof for the generalization guarantee upper-bound for the problem of learning to warm-start Lagrangian relaxation.

*Proof of Theorem 6.1.* The proof simply relies on the closed-form solution of the ERM estimator for the squared loss. First, given an initialization $\phi \in \Pi$, the risk $R(\phi) = \mathbb{E}_{P \sim \mathcal{D}}[\|\phi - \pi^*(P)\|_2^2]$. Therefore, given a problem distribution $\mathcal{D}$, the optimal initialization $\phi^*(\mathcal{D})$ corresponding to $\mathcal{D}$ is simply $\phi^*(\mathcal{D}) = \mathbb{E}_{P \sim \mathcal{D}}[\pi^*(P)]$. Therefore, the excessive can be written as

$$\mathbb{E}_{P \sim \mathcal{D}}[\|\hat{\phi}(S) - \pi^*(P)\|_2^2] - \mathbb{E}_{P \sim \mathcal{D}}[\|\pi^*(\mathcal{D}) - \pi^*(P)\|] = \|\hat{\phi}(S) - \phi^*\|_2^2.$$

Let $Z_i = \pi^*(P_i)$, and from Assumption 4.2, the $s$-dimensional random vector $Z_i$ are i.i.d. with mean $\phi^*(\mathcal{D})$ and belongs to the bounded domain $Z_i \in [0, \pi_{\max}]^s$ due to Assumption 4.2. Therefore, applying the standard Popoviciu's inequality on variance (Lemma B.3), we have

$$\mathbb{E}_{P \sim \mathcal{D}} = \frac{1}{N}\sum_{i=1}^N \sum_{k=1}^s \text{Var}(Z_{i,k}) \leq \frac{s\pi_{\max}^2}{4N} = \mathcal{O}\left(\frac{s}{N}\right),$$

where $Z_{i,k}$ is the $k$-th coordinate of $Z_i$. □

## D.2. Lower Bound

We now present a detailed proof of the minimax lower bound for the problem of learning to warm-start Lagrangian relaxation.

**Theorem 6.2 (restated).** *For any learning algorithm $\hat{\phi}$ taking input as a set $S$ of $N$ problem instances $P_1, \ldots, P_N \sim \mathcal{D}$, the minimax expected risk is lower-bounded by*

$$\inf_{\phi} \sup_{\mathcal{D} \in \Delta(\mathcal{P})} \mathbb{E}_{S \sim \mathcal{D}^N}\left[\mathbb{E}_{P \sim \mathcal{D}}[\|\phi(S) - \pi^*(P)\|_2^2] - \mathbb{E}_{P \sim \mathcal{D}}[\|\phi^* - \pi^*(P)\|_2^2]\right] = \Omega\left(\frac{s}{N}\right).$$

*Proof of Theorem 6.2.* Similar to Theorem 5.6, our approach is to leverage Fano's method (Theorem 3.3). However, since the loss here is quadratic, the overall proof can be simplified drastically. We proceed with the following steps.

**Step 1: Construction of hard instances.** In this construction, we also leverage the idea from the previous case (Theorem 5.6), by restricting the problem instance to take only the form $P = (c, \mathbf{I}_s, \frac{1}{2} \cdot \mathbf{1}_s, 0_{t \times s}, 0_t)$. From Lemma 5.7, we show that problem instance $P$ of this form has the property $u(\pi, P) = \sum_{k=1}^{s} \min(\frac{\pi_k}{2}, c_k - \frac{\pi_k}{2})$. This function is strictly concave and is maximized uniquely when $\frac{\pi_k}{2} = c_k - \frac{\pi_k}{2}$, if $c_k > 0$ since the Lagrangian multiplier has to be positive. Therefore, for any $c$ such that $c_k \geq 0$ for all $k$, we have $\pi^*(P) = c$.

**Step 2: Family of distribution construction.** We define a family of distributions for $c$, parameterized by a binary vector $v \in \{0, 1\}^s$ and a small perturbation scale $\epsilon \in (0, 1/2)$ that we will choose later.

- Support: for any $k$, $c_k$ takes value in $\{1, 2\}$.

- Probability: the probabilities are controlled by $v$ and $\epsilon$ as follows

  - If $v_k = 1$: $\mathbb{P}(c_m = 2) = \frac{1+\epsilon}{2}$ and $\mathbb{P}(c_k = 1) = \frac{1-\epsilon}{2}$.
  - If $v_k = 0$: $\mathbb{P}(c_k = 2) = \frac{1+\epsilon}{2}$ and $\mathbb{P}(c_k = 1) = \frac{1+\epsilon}{2}$.

Then, for a problem distribution $\mathcal{D}_v$ constructed this way, we have $(\phi^*(\mathcal{D}_v))_k = \mathbb{E}[c_k] = \frac{3}{2} + \frac{\epsilon}{2}(2v_k - 1)$

**Step 4: Minimax lower-bound.** From Varshamov-Gilbert bound in Lemma C.1, there is a packing set $\mathcal{V} = \{v^{(1)}, \dots, v^{(M)}\} \subset \{0, 1\}^s$ with $M \geq 2^{s/8}$ such that $d_H(v^{(i)}, v^{(j)}) \geq s/8$. Therefore, for any two $v^{(i)}, v^{(j)} \in \mathcal{V}$, we have

$$\|\phi^*(\mathcal{D}_{v^{(j)}}) - \phi^*(\mathcal{D}_{v^{(i)}})\|_2^2 = \sum_{k=1}^{s} \left( \frac{\epsilon}{2}(2v_k^{(j)}) - 2v_k^{(i)} \right)^2 = \epsilon^2 \cdot d_H(v^{(j)}, v^{(i)}) \geq \frac{\epsilon^2 s}{8}. \tag{5}$$

Again, we note that the KL divergence for the Bernoulli distributions with parameter $p, q \in [\frac{1-\epsilon}{2}, \frac{1+\epsilon}{2}]$ satisfies $\mathrm{KL}(p \| q) \leq \frac{(p-q)^2}{q(1-q)} \leq 4\epsilon^2$. Therefore,

$$\mathrm{KL}(\mathcal{D}_{v^{(i)}}^N \| \mathcal{D}_{v^{(j)}}^N) \leq 4N \sum_{k=1}^{s} (p_k^{(j)} - p_k^{(i)})^2 = 4N\epsilon^2 d_H(v^{(i)}, v^{(j)}) \leq 4Ns\epsilon^2.$$

We choose $\epsilon$ such that

$$4Ns\epsilon^2 \leq \frac{s}{16} \log 2 \Rightarrow \epsilon \asymp \frac{1}{\sqrt{N}}.$$

Substituting to (5), we conclude that

$$\inf_{\phi} \sup_{\mathcal{D} \in \Delta(\mathcal{P})} \mathbb{E}_{S \sim \mathcal{D}^N} \left[ \mathbb{E}_{P \sim \mathcal{D}}[\|\phi(S) - \pi^*(P)\|_2^2] - \mathbb{E}_{P \sim \mathcal{D}}[\|\phi^* - \pi^*(P)\|_2^2] \right] = \Omega \left( \frac{s}{N} \right).$$

The proof is complete. $\square$

