# OpenReview forum: "Provably Data-driven Lagrangian Relaxation for Mixed Integer Linear Programming"
_ICML.cc/2026/Conference — ICML 2026 regular_

### Official Review · Reviewer_niA9 · 2026-02-21

**Soundness:** 2
**Presentation:** 2
**Significance:** 3
**Originality:** 3
**Overall Recommendation:** 4
**Confidence:** 3

**Summary:**

This paper studies data-driven selection of Lagrangian multipliers for Lagrangian relaxation (LR) of MILPs with $s$ coupling constraints, assuming instances are drawn i.i.d.\ from a distribution $D$.
Under the restricted search domain $\Pi$ (Assumption 4.2), it analyzes the ERM multiplier estimator via expected excess risk and proves an upper bound of order $O(s^{1.5}/\sqrt{N})$ using a standard Rademacher-complexity pipeline.
It also proves an information-theoretic minimax lower bound of order $\Omega(s/\sqrt{N})$ via a hard family of distributions and Fano's method, implying linear dependence on $s$ is unavoidable in the worst case.
Finally, it studies a learning-to-warm-start objective based on squared distance to instance-optimal multipliers and shows a minimax-optimal fast rate $\Theta(s/N)$.

**Compliance With Llm Reviewing Policy:**

Affirmed.

**Final Justification:**

The paper makes a meaningful theoretical contribution by providing nontrivial upper and lower bounds for data-driven learning of Lagrangian multipliers, together with a clean and tight $\Theta(s/N)$ result for the warm-start setting on an important problem.

The rebuttal resolved my main clarification questions about the interpretation of Theorem 5.6 and the Section 6 estimator/notation, and the follow-up explanation makes the proposed SGA-with-averaging improvement seem like a plausible and natural extension of the current analysis. I am raising my score to 4 (Weak Accept).

**Key Questions For Authors:**

1. Lower bound (Theorem 5.6) vs. linear-class Rademacher complexity.
The paper proves a minimax lower bound for direct multiplier learning:
$$
\inf_{\hat\pi}\sup_{D\in\Delta(\mathcal P)} \mathcal E(\hat\pi(S))
= \Omega\left(\frac{s}{\sqrt{N}}\right).
$$
Later, the authors remark that for purely linear function classes, the Rademacher complexity (and hence typical generalization rates) scale as
$O\left(\frac{s}{\sqrt{N}}\right)$.

a. What is the precise connection between the hard-instance family used in Theorem 5.6 and a linear (or ``essentially linear'') function class, and in what sense do they induce the same $\frac{s}{\sqrt{N}}$ scaling?

b. Since the dual-objective class $u(\pi,P)$ is more structured (piecewise-linear concave, optimization-defined) than a purely linear class, should we expect a strictly larger minimax rate than linear prediction, and if so, does Theorem 5.6 mainly reflect only a ``linear subclass'' lower bound?

2. Notation and estimator in Section 6: what is $\hat{\phi}$?
Section 6 defines the warm-start loss (up to notation):
$$
\ell(\phi,P) := ||\phi - \pi^\ast(P)||_2^2,
$$
and studies the ERM estimator $\hat{\phi}(S)$.

a. Is $\hat{\phi}(S)$ intended to be the empirical mean $\frac{1}{N}\sum_{i=1}^N \pi^\ast(P_i)$ (possibly projected onto $\Pi$), or is it meant to coincide with the earlier direct-learning ERM $\hat{\pi}(S)$ from Section 4?

b. The proof sketch for Theorem 6.1 states $\hat{\pi}(S)$ is simply the sample mean; is this a typographical mismatch that should read $\hat{\phi}(S)$?

**Limitations:**

Yes.

**Strengths And Weaknesses:**

Strengths:

1, Generalization analysis for LR multipliers with matching upper/lower guarantees up to a $\sqrt{s}$ factor.

The paper focuses on the generalization behavior of learning Lagrangian multipliers for MILPs, which is a practically important component of LR-based decomposition and branch-and-bound pruning.
On the upper-bound side, it derives an explicit Rademacher-complexity bound for the LR dual-value function class by exploiting the concavity of $u(\pi,P)$ in $\pi$ and a uniform bound on (super)subgradients, which yields Lipschitzness and enables a classic covering-number/Dudley analysis.
On the lower-bound side, it constructs a family of hard instances/distributions and establishes a minimax lower bound via Fano's method, clarifying that a linear dependence on the number of coupling constraints $s$ is information-theoretically unavoidable.

2, Warm-start learning admits a tight fast rate with a simple interpretation.

For the warm-start loss $\ell(\varphi,P)=\|\varphi-\pi^\ast(P)\|_2^2$, the paper provides both an upper bound and a matching minimax lower bound, yielding a tight rate $\Theta(s/N)$.
The analysis is clean and interpretable: ERM for squared loss reduces to estimating the mean of $\pi^\ast(P)$, so the estimator is essentially the sample mean and the excess risk becomes a mean-estimation error, which explains the statistical advantage of warm-starting compared to direct dual maximization.


Weaknesses:

1, Upper bound leaves a $\sqrt{s}$ gap and does not exploit piecewise-linear structure.

As discussed in the paper, the direct-learning upper bound $O(s^{1.5}/\sqrt{N})$ does not match the minimax lower bound $\Omega(s/\sqrt{N})$, leaving a $\sqrt{s}$ gap.
The authors attribute this to the generic Lipschitz/covering-number analysis not exploiting the piecewise-linear (polyhedral) structure of the Lagrangian dual function.

2, Notation/presentation issues in the draft.
Section~6 contains a notational mismatch in the proof sketch (referring to $\hat\pi(S)$ as the sample mean while discussing the warm-start ERM estimator), which could confuse readers.

---

> ### Author Rebuttal · Authors · 2026-03-31
>
> We thank the reviewer for the very detailed feedback. We will address the reviewer’s concerns as follows.
>
> ### Questions
>
> 1a1. Q. In what sense do the linear (or essentially ) function classes induce the same $\mathcal{O}(s/\sqrt{N})$ expected excess risk rate?
>
>    A. As discussed with Reviewer ekfU, concretely, for a linear function class $\\mathcal{F} = \\{f_w: [-1, 1]^s \\rightarrow \\mathbb{R} \\mid w \\in [-1, 1]^s]\\}$, we can show that $\\mathscr{R}_N(\\mathcal{F}) = \\mathcal{O}(s/\\sqrt{N})$.
>
>    Subsequently, we can show that for a piecewise linear function class of $K$ pieces, i.e. $\\mathcal{F} = \\{f_W: [-1, 1]^s \\rightarrow R \\mid W \\in [-1, 1]^{K \\times s}\\}$, where  $W = \\{w_1, \\dots, w_K\\}$, the Rademacher complexity $\\mathscr{R}_N(\\mathcal{F}) = \\mathcal{O}(Ks/\\sqrt{N})$.  Note that in the latter case, the UPPER-BOUND scales with $K$, which can scale exponentially with $s$.
>
> 1a2. Q: Relation with our lower-bound construction in Theorem 5.6
>
>    A: That is a very good catch! Though the function $u(\\pi, P)$ is defined via an optimization problem, we construct the hard family of problem distributions carefully so that for any problem instances $P$ that come from that family, $u(\\pi, P)$ can be written as the sum of $s$ one-dimensional piecewise linear functions (each has 5 pieces, independent on $s$).
>
> 1a3. Q:  In what sense do they have the same rate $s/\sqrt{N}$
>
>   A: We note that the rate $s/\sqrt{N}$ in 1a1 is an excess risk upper-bound, which is for the linear function class only, and derived using Rademacher complexity, while the rate in 1a2 is the excess risk lower-bound, applying to the function class $\mathcal{U} = \{u_\pi: \mathcal{P} \rightarrow [0, H] \mid \pi \in \Pi\}$, generally defined via an optimization problem. Though appearing to have the same factor $s/\sqrt{N}$, they are different objects (lower-bound vs. the upper-bound), applying to different function classes (simple linear function class vs. a complex function class defined via an optimization problem).
>
> 1b1.  Q: Since $u(\pi, P)$ is more structured, should we expect a strictly larger minimax rate on that?
>
>   A: Not really, __as discussed with Reviewer ekfU (see above)__, we provide a learning algorithm (SGA with averaging instead of ERM learner) that can achieve the minimax optimal rate of $\Theta(s/\sqrt{N})$ exactly. We will incorporate this discussion on the revised draft.
>
> 1b2. Q:  Does Theorem 5.6 mainly reflect only a linear subclass lower-bound?
>
>   A: Not really, Theorem 5.6 actually captures the fundamental limit of the entire learning Lagrangian relaxation problem. In our construction, we are not constructing linear function classes, but constructing a specific hard family of distributions over MILP problem instances, of which each function $u(\pi, P)$ is defined via an optimization, as noticed by the Reviewer. We then use an information-theoretic technique (Fano’s) to show that for those problem distributions, no matter which learning algorithm you use, you cannot achieve a better rate of $\Omega(s/\sqrt{N})$. This establishes the fundamental limit for the problem of learning the Lagrangian relaxation itself.  __We will elaborate on this point more to avoid confusion on the revised draft.__
>
> 2a. Q: Is $\hat{\phi}(S)$ intended to be $\frac{1}{N}\sum_{i = 1}\pi^*(P_i)$ (possibly projected onto $\Pi$)?
>
>   A. Your understanding is correct on this point
>
> 2b. Q: Typo on Proof sketch for Theorem 6.1.
>
>   A. Thank you for catching this. We will fix it in the revised draft.
>
> ### Other clarifications
> 1. “Upper-bound leaves a $\sqrt{s}$ gap and does not exploit piecewise linear structure”: at the point of the submission, we hypothesized that leveraging the piecewise structure of $u(P, \pi)$ can potentially close the $\sqrt{s}$ gap. However, as discussed with you and the Reviewer ekfU, leveraging this structure only helps if the number of pieces $K$ is independent of $s$, but in our case, $K$ can scale exponentially with $s$, which __makes the situation worse__. Moreover, we provided other learning algorithms (SGA with averaging instead of the ERM learner) that can achieve minimax optimal rate in learning Lagrangian relaxation, which __closes the  $\\sqrt{s}$ gap without invoking the piecewise linear structure__ at all. We will incorporate this discussion in the revised draft.
>
> ### Conclusions
> We hope that our rebuttal addresses the reviewer’s concern. We are happy to address further questions if needed. __We respectfully request that the reviewer re-evaluates the work in light of our rebuttal__. Many thanks!

---

> > ### Author Rebuttal · Reviewer_niA9 · 2026-04-04
> >
> > I thank the authors for the helpful rebuttal. It clarifies the relation between the hard-instance construction in Theorem 5.6 and linear/piecewise-linear classes, makes clear that the lower bound is intended for the learning Lagrangian relaxation problem itself rather than only a linear subclass, and resolves my Section 6 question by confirming that the warm-start estimator is the empirical mean and that the use of $\hat{\pi}(S)$ there was a typo.
> >
> > However, the rebuttal suggests that an SGA-with-averaging learner may achieve the optimal $\Theta(s/\sqrt{N})$ rate, but this is new material that is not part of the submitted paper, which requires an extra work and has significant difference with the current submission.

---

> > > ### Author Response · Authors · 2026-04-04
> > >
> > > We thank the reviewer for the prompt response. We are glad __the reviewer finds that our rebuttal addresses all of their technical concerns__.
> > >
> > > Regarding your valid point that the SGA-with-averaging result is "new material", we completely understand your perspective. However, we would like to respectfully offer __some context for you (and the AC) as to why we view this result as a natural and seamless completion of the paper's core narrative__, rather than a significant departure from it.
> > >
> > > 1. __Completing the theoretical picture__: A central contribution of our draft is the information-theoretic minimax lower-bound of $\Omega(s/\sqrt{N})$, which __applies to ANY learning algorithm__. Since our upper-bound for ERM left a $\sqrt{s}$ gap, a very __natural theoretical question arose: is our universal lower-bound loose, or is there any other algorithm that can achieve that lower-bound?__
> > >
> > > By demonstrating that the SGA-with-averaging learner achieves the $\mathcal{O}(s/\sqrt{N})$ rate, __we constructively answer this question__. This completes the theoretical story we set out to tell.
> > >
> > > 2. __Complementing, not replacing, the ERM analysis__: The SGA result does not render our original ERM analysis obsolete. ERM remains the most fundamental statistical learning paradigm and is widely adopted in practice. Characterizing its specific generalization guarantee is independently valuable. The __SGA analysis simply enriches the paper by providing practitioners with an optimal algorithmic counterpart__.
> > >
> > > 3. __Seamless integration with "minimal extra work"__: While the theoretical implications of closing the gap are significant, the mathematical "extra work" is rather minimal. Concretely, the full proof for SGA is just around a page and relies on standard stochastic optimization techniques and learning theory (online-to-batch conversion), __leveraging the exact same geometric properties of MILP presented in Proposition 5.1 (already established in the original draft)__. This means that __the new result only requires a short new subsection and does not disrupt the paper's existing structure and narrative.__
> > >
> > > 4. __A direct result of the rebuttal__: We view __closing the $\sqrt{s}$ gap as a direct result of our discussions with you__ (and with Reviewer ekfU). We believe that __providing a theoretical resolution to a gap identified by a reviewer is exactly in the spirit of the interactive rebuttal phase.__
> > >
> > > __While we hope you might consider re-evaluating our work based on the paper's improvements, we fully respect your view__. Thank you for your time, engagement, and valuable feedback. Have a good day!

---

### Official Review · Reviewer_ekfU · 2026-03-11

**Soundness:** 3
**Presentation:** 3
**Significance:** 2
**Originality:** 2
**Overall Recommendation:** 3
**Confidence:** 3

**Summary:**

This paper studies learning Lagrangian multipliers from data for mixed integer linear programming. It gives an upper bound of $O(s^{1.5}/\sqrt{N})$ for the ERM multiplier, a lower bound of $\Omega(s/\sqrt{N})$, and, in the learning-to-warm-start setting, a minimax optimal rate of $\Theta(s/N)$. The analysis uses Rademacher complexity for the upper bound and Fano's method for the lower bound.

**Compliance With Llm Reviewing Policy:**

Affirmed.

**Final Justification:**

The authors’ responses have addressed my main concerns and I appreciated their hard working during the rebuttal phase however I choose to maintain the score.

**Key Questions For Authors:**

1. Can the $\sqrt{s}$ gap be closed? Have you attempted analyses that exploit the piecewise-linear structure of $u(\pi, P)$ rather than generic Lipschitz properties?
2. For the instances studied by Demelas et al. (2024), what are typical values of $s$, $B$, and $\pi_{\max}$? Are the resulting bounds informative in practice?
3. Is the warm-start formulation's advantage robust when the predictor is restricted to a parametric family rather than all of $\Pi$?

**Limitations:**

Yes

**Strengths And Weaknesses:**

### Strengths

1. Clear and well-motivated problem. The paper addresses a genuine theoretical gap: Demelas et al. (2024) demonstrated empirical success of learning Lagrangian multipliers, but no sample complexity analysis existed.
2. The warm-start result is the paper's strongest contribution. Theorem 6.1 and Theorem 6.2 establish that learning to warm-start achieves a minimax optimal rate of $\Theta(s/N)$, compared to $\Omega(s/\sqrt{N})$ for direct dual maximization.
3. Correct and complete proofs. The lower-bound construction is carefully executed, and the coordinate-wise decomposition of the dual function works cleanly in this setting.

### Weaknesses

1. The $\sqrt{s}$ gap in the direct learning bounds weakens the main result. The upper bound $O(s^{1.5}/\sqrt{N})$ and lower bound $\Omega(s/\sqrt{N})$ differ by $\sqrt{s}$, so the true rate for direct learning remains unresolved.
2. Technical novelty is moderate. The proof techniques build on standard tools, and the contribution is more in adapting known tools to this problem than in introducing new analytical techniques.
3. No experiments. The practical relevance of the bounds is unclear, and it is not verified whether the $\sqrt{s}$ gap in the upper bound manifests in practice.
4. Assumptions may be restrictive. Assumption 4.1 can make the Lipschitz constant $L = 2B\sqrt{s}$ large, and Assumption 4.2 makes the bound depend directly on $\pi_{\max}$.

---

> ### Author Rebuttal · Authors · 2026-03-31
>
> We thank the reviewer for the constructive feedback. We are glad that the reviewer found our paper clear and well-motivated. We will address the reviewer’s concerns as follows
>
> ### Questions
> 1. Q. Attempt to exploit the piecewise affine structure of $u(\pi, P)$
>
>     A. Yes. At the point of submission, we hypothesize that we can improve the expected excess risk of the ERM learner by using the piecewise linear structure of $u(\pi, P)$. But after investing in this direction, we found out that leveraging this structure might lead to more issues rather than closing the gap.
>
>     To see that, we have to look at the Rademacher complexity of the piecewise linear function class. Concretely, we can show that for a piecewise linear function class of $K$ pieces, i.e. $\mathcal{F} = \{f_W: [-1, 1]^s \rightarrow R \mid W \in [-1, 1]^{K \times s}\}$,
>
>      where  $W = \\{w_1, \dots, w_K\\}$, and $f_W(\\pi) = \\max_{k = 1}^K  w_k^\\top \\pi$  the Rademacher complexity $\mathscr{R}_N(\mathcal{F}) = \mathcal{O}(Ks/\sqrt{N})$. Ostensibly, this gave us a feeling that maybe if we could control $K$ nicely, our bound would scale with $s/\sqrt{N}$ only.
>
>      However, it is not the case. In fact, $K$ scales badly with the dimension of the problem instance. For example, consider the following simple MILP: we only have $s$ binary variables $x \\in \\{0, 1\\}^s$, the objective is $\\min_{x} \\mathbf{1}_s^\\top x$, the coupling constraints are
>
>      $x_i \geq 1/2$, and the domain of Lagrangian is $\Pi = [0, 2]^s$. The Lagrangian dual function is $u(\pi, P) = \min_{x \in \{0, 1\}^s} \sum_{i = 1}^s [x_i + \pi_i(1/2 - x_i)] = \sum_{i = 1}^s \min(1/2 \pi_i, 1 - 1/2 \pi_i)$. For each coordinate $i$, the active piece has a kink when $\pi_i = 1$. Because there are $s$ independent coordinates, these $s$ hyperplanes slice parameter space into exactly $K = 2^{s}$ distinct orthants. Thus, $u(\pi, P)$ is piecewise linear in $K = 2^{s}$ pieces.
>
>     Moreover, later we found out that __the piecewise linear structure is not required for closing the $\sqrt{s}$__, but instead via algorithmic stability (Q2 below for discussion). We will remove the hypothesis, as well as update this discussion on the revised draft.
>
> 2. Q: Can the  $\sqrt{n}$ gap be closed?
>
>     A:  Thank you for the question. Fortunately, the answer is __yes__. Concretely, we provide an algorithm (SGA with averaging instead of the ERM learner that we investigated in Theorem 5.5) that can achieve the optimal minimax rate of $\\Theta(s/ \\sqrt{N})$, therefore closing the gap.
>
>      Concretely, instead of using ERM learner as $\\hat{\\pi}(S) \\in \\arg \\min_{\\pi} \\sum_{i  =1}^N u(\\pi, P_i)$, we instead using SGA with averaging as follows
>
>      (1) Intialize $\pi_1 = 0$.
>
>      (2)  For $t = 1, \\dots, N$ do: (2.1) get $P_t$, (2.2) solve Lagrangina sub-problem to get $x^*_t \\in \\arg \max_{x}  u(\\pi_t, P_t)$
>
>      (2.3) calculate the sub-gradient $g_t = b_t - A_tx^*_t$, and
>
>      (2.4) Update and projecting $\pi_{t + 1} = \textup{Proj}_{\Pi}(\pi_t - \eta g_t)$.
>
>       (3) Finally, output $\\bar{\\pi}_N = \\frac{1}{N} \\sum_{t = 1}^N \\pi_t$
>
>       We can show that if setting $\\eta = \\pi_{max}/(2B\\sqrt{N})$, the algorithm above will achieve the minimax optimal rate of $\Theta(s/\sqrt{N})$. __We will incorporate this change (with a formal theorem and proof) in the revised draft.__
>
> 3. Q:  If the assumptions are restrictive, i.e., the upper-bound might scale with the constants $B$ and $\\pi_{max}$ in the assumptions.
>
>     A: That is a good catch! Actually, the value $B$ and $\\pi_{max}$ can appear in the lower-bound as well, with exactly the same factor $B  \\pi_{max}$ as in the upper-bound. To do that, we just have to slightly modify the lower-bound construction (Theorem 5.6) a bit. Concretely, instead of considering the coupling matrix $A = \\mathbf{I}_s$, the constraint vector $b = \mathbf{1}_s$, and the component of the objective vector $c_i \in \{0, 1\}$ in the original construction, we can scale it to $A = B \mathbf{I}_s$, $b = \frac{B}{2}  \mathbf{1}_s$, and choose
>
>      $c_i \in \\{0, B\\pi_{max}\\}$. The rest of the proof remains the same (with some small adjustment adapting to the new values), and the lower-bound will have the factor $B \pi_{max}$ as in the upper-bound. We will provide this modification in the revised draft.
>
> 4. On the novelty: Content-wise, our contribution is drawing a connection between empirical success of learning-to-optimize approaches and the theoretical foundation, which is under-investigated. Technical-wise, our lower bound constructions are novel in the literature. We hope that it serve as a starting point for subsequent works.
>
> ### Summary
> We hope that our rebuttal addresses the reviewer’s concern. We are happy to address further questions if needed. __We respectfully request that the reviewer re-evaluates the work in light of our rebuttal__. Many thanks!

---

> > ### Author Rebuttal · Reviewer_ekfU · 2026-04-02
> >
> > My concerns have been addressed. But I still maintain the score.

---

### Official Review · Reviewer_FZFT · 2026-03-12

**Soundness:** 4
**Presentation:** 4
**Significance:** 3
**Originality:** 4
**Overall Recommendation:** 5
**Confidence:** 3

**Summary:**

The paper provides a theoretical explanation for the empirical success of the learning-to-optimize approach for selecting multipliers for the Lagrangian relaxation (LR) of a mixed-integer linear program (MILP). The paper casts the multiplier selection problem as a statistical learning model and the main theoretical results include (1) a generalization bound for selecting the Lagrangian multipliers by solving an empirical LR of a group of training MILP instances and (2) a generalization bound for the Euclidean distance between the training-optimal Lagrangian multipliers and the true-optimal counterpart.

**Compliance With Llm Reviewing Policy:**

Affirmed.

**Final Justification:**

The rebuttal has reinforced my prior assessments, which are (1) the theoretical results appear a bit disconnected with the empirical success (that is, [1]) that motivates the study, but (2) the perspective and the theoretical results on their own are novel and provide inspiration for subsequent studies. For this reason, I maintain my positive overall recommendation (5: Accept).

[1] Demelas, F., Le Roux, J., Lacroix, M., and Parmentier, A. Predicting Lagrangian multipliers for mixed integer linear programs. In Forty-first International Conference on Machine Learning, 2024.

**Key Questions For Authors:**

1. The theoretical results appear disconnected with the empirical success (that is, [1]) that motivates the study. Specifically, [1] studies a learning-to-select approach that tailors to individual problem instances (that is, a mapping from instance P to multiplier \pi). In contrast, the paper intends to select an individual multiplier \pi that works for a whole distribution of instances. Can you generalize the results for adaptive multiplier selection?
2. In Section 6, the uniqueness assumption on the optimal multiplier \pi^*(P) appears to be missing. If I understood correctly, you intended to assume that there exists a unique \pi^*(P) that is optimal to the Lagrangian relaxation of an MILP. However, based on my experience, this assumption is typically not satisfied (and computationally challenging to check) for MILPs. Can you generalize the results (that is, Theorems 6.1 & 6.2) to the case of multiple optimal multipliers? In that case, the squared Euclidean distance needs to be revised to be the distance from the initial multiplier \phi to the set of optimal multipliers.

[1] Demelas, F., Le Roux, J., Lacroix, M., and Parmentier, A. Predicting Lagrangian multipliers for mixed integer linear programs. In Forty-first International Conference on Machine Learning, 2024.

**Limitations:**

Yes. The impact statement appears to be missing though.

**Strengths And Weaknesses:**

Soundness: the main theoretical results and the proofs make sense to me.

Presentation: the paper is carefully written and well articulated.

Significance: the techniques largely follow the established statistical learning theory literature, but the theoretical results are still relevant in the sense that it may serve as the starting point for subsequent works.

Originality: I appreciate the study of the theoretical foundation for the empirical success of learning-to-optimize approaches.

---

> ### Author Rebuttal · Authors · 2026-03-31
>
> We thank the reviewer for the positive and constructive feedback. We are glad that the reviewer appreciates our connection between empirical success and the statistical foundation of the learning-to-optimize approaches. We will address the reviewer’s concern as follows.
>
> ### Questions
> 1. Q: Removing the uniqueness assumptions on the learning-to-initialize setting
>
>     A: That is a very good catch, and thank you for suggesting the point-to-set distance potential fix. However, we afraid that using definition might cause the “moving target” issue, and make the upper-bound analysis problematic. Concretely, if we define $\\pi^*(P) = \\arg \min_{\\pi} \\|\\pi - \\phi\\|_2^2$, where $\\phi$ is our initialization and define
>
>    $\\ell(\\phi, P) = \\|\\phi - \\pi^\\star(P)\\|_2^2$ then  $\\pi^\\star(P)$ will be depending on $\phi$, and break the strongly convex property of $\ell$ w.r.t. $\phi$.
>
>    However, there is a much simpler fix, which is using a breaking tie arbitrarily but consistently when choosing $\\pi^\\star(P) \in \\Pi^\star(P)$. Concretely, one can simply use the minimum norm definition $\\pi^\\star(P) = \\arg \\min_{\\pi \in \\Pi^\\star(P)} \\| \\pi \\|_2$, which is exactly
>
>    $Proj_{\\Pi^\\star(P)}(0_p)$, where $Proj_{\\Pi^\\star(P)}(0)$
>
>     is the projection of the point $0$ onto the closed and convex solution set $\\Pi^\\star(P)$ (cause it is a solution set of a convex optimization problem with a closed feasible set and continuous objective), which is guaranteed to be unique due to the Hilbert projection theorem. By doing so, the proof of the upper-bound remains exactly the same, and the lower-bound also remains unchanged because the set $\\Pi^\\star(P)$ is already unique due to our construction.
>
>     __We thank the reviewer for the great suggestion and will modify our Section 6 accordingly.__
>
> 2. Q: Drawing the connection to learning the mapping with the parametric family.
>
>     A: Another very good catch! Though our focus is to provide a base case (static parameters) to rigorously characterize the sample complexity of learning the Lagrangian multipliers, we can definitely provide guarantees for learning the mapping $f_\theta(P) = \pi(P)$, where $f_\theta$ is an MLP parameterized by $\theta$ in the following settings.
>
>     Concretely, let $\\mathcal{F} =  \\{f_\\theta: \\mathcal{P} \\rightarrow \\Pi \\mid \\theta \\in \\Theta \\}$ is the class of NNs parameterized by the weights $\\theta$. First, in our analysis, we established that $u(\\pi, P)$ is $L$-Lipschitz w.r.t $\\pi$ in the $\\ell_2$-norm (with $L = \\mathcal{O}(B\sqrt{s})$).  Second, let $\\mathcal{U}_{\\mathcal{F}}=$
>
>      $\\{P \\rightarrow u(f_\\theta(P), P)\\}$  then we can bound the $\\hat{ \\mathscr{R}}  (\\mathcal{U}_{\\mathcal{F}})$
>
>       $ \leq \\sqrt{2}L\\sum_{j = 1}^s \hat{ \\mathscr{R}}   (\\mathcal{F}_j)$ by Maurer’s Vector Contraction Inequality [1], where $\\mathcal{F}_j$ is the hypothesis class of the $j^{th}$ output neuron. And finally, we can use some size-independent generalization bounds (e.g., [2]) to bound $\\mathscr{R}(\\mathcal{F}_j)$, for example $\\hat{\\mathscr{R}}(\\mathcal{F}_j)$
>
>      $\leq \mathcal{O}\left( \frac{\sqrt{d} B_x}{\sqrt{N}} \prod_{k=1}^d \\|W_k\\|_F \right)$,
>
>       where $W_k$ is the matrix weights of the $k^{th}$ layer of the MLPs $f_\\theta$. We will incorporate this discussion in our revised draft.
>
> ### Other clarifications
> 1. Resolving the $O(\\sqrt{s})$ gaps: as discussed with Reviewers ekfU and niA9, __we closed the gap of $O(\\sqrt{s})$__ to our information-theoretic lower-bound (for any learning algorithm) by providing an algorithm (SGA with averaging) that can match the rate. See the discussion with Reviewers ekfU and niA9 below for details. We will incorporate this change in the revised draft.
>
> 2. Clarification on the novelty: on the content side, we are glad that the reviewer appreciates our connection between the empirical success of learning-to-optimize approaches and the theoretical foundation, which is largely underinvestigated in the literature. On the technical side, we note that our lower-bound constructions are novel in the literature. We hope that this work may serve as a starting point for subsequent works.
>
> ### Conclusion
> We hope that our answers address the reviewer’s concerns, and we are happy to answer any further questions from the reviewer. Many thanks!
>
> ### References
> [1] Andreas Maurer, A vector-contraction inequality for Rademacher complexities, ALT’16
>
> [2] Golowich et al. Size-Independent Sample Complexity of Neural Networks, COLT’18

---

> > ### Author Rebuttal · Reviewer_FZFT · 2026-04-02
> >
> > Thanks for offering the responses!
> >
> > 1. I was confused why the same proofs would work for replacing the unique \pi^* with the minimum-norm \pi^*. Please elaborate more.
> >
> > For example, does the new theoretical guarantee assume that the empirical  optimal multiplier \hat{\phi}(S) to be unique, or does it require \hat{\phi}(S) to be the minimum-norm optimal multiplier?
> >
> > 2. The proposed instance-wise guarantees for neural nets sound very different from the existing results, in terms of both the claim and the rate. I raised this question in the hope that the same analyses may be extended to handle this more relevant case, but it seemed not. At this point, I would suggest keep the paper clean and consistent without mentioning these instance-wise guarantees.

---

> > > ### Author Response · Authors · 2026-04-02
> > >
> > > We thank the reviewer for the following follow-up question.
> > >
> > > 1. " Does the new theoretical guarantee assume that the empirical  optimal multiplier $\hat{\phi}(S)$ is unique?": Actually, $\hat{\phi}(S)$ is unique as long as $\pi^\star(P_i)$, for $P_i \in S$, is uniquely defined (like the proposed minimum-norm definition, or more generally, "break-tie arbitrarily but consistently definition"). In such a case, $\hat{\phi}(S)$ simply minimizes $\\sum_{i = 1}^N \\|\\phi - \\pi^\\star(P_i)\\|_2^2$,  which is exactly
> > >
> > >      $\hat{\phi}(S) = \frac{1}{N}\sum_{i = 1}^N \pi^\star(P_i)$. Hence, it does not require that $\hat{\phi}(S)$ be the minimum-norm, or we do not need to assume uniqueness for it, because it is mathematically guaranteed by definition. __We will elaborate this point in clear detail in the revised draft to avoid confusion.__
> > >
> > > 2. "Excluding the instance-wise guarantees": thank you for the suggestion! We agree that adding the results requires changing the narratives dramatically, and keeping this paper this way with added results for a distributional setting would make the paper much clearer. We will exclude that part in the revised draft.
> > >
> > > Should the reviewer still have any concerns, feel free to let us know. Many thanks!

---

### Official Review · Reviewer_haky · 2026-03-20

**Soundness:** 3
**Presentation:** 3
**Significance:** 3
**Originality:** 3
**Overall Recommendation:** 4
**Confidence:** 1

**Summary:**

This paper addresses the lack of theoretical guarantees for data-driven Lagrangian Relaxation (LR) in MILP solving, formalizing multiplier learning as a statistical problem. It derives key results: an upper bound of $\(O(s^{1.5} / \sqrt{N})\)$ and minimax lower bound of $\(\Omega(s / \sqrt{N})\)$ for learned multipliers, plus a minimax-optimal $\(\Theta(s / N)\)$ rate for learning-to-warm-start.

**Compliance With Llm Reviewing Policy:**

Affirmed.

**Key Questions For Authors:**

I acknowledge insufficient background in Lagrangian Relaxation, which prevents me from fully understanding the paper’s technical derivations and validating its theoretical rigor. I am not capable of making an accurate, detailed assessment of the work’s correctness or practical relevance, and thus defer to reviewers with specialized expertise in this domain.

**Strengths And Weaknesses:**

- Strengths: Theoretically rigorous, with sound problem formulation and novel sample complexity bounds; well-structured, with clear connections to real-world MILP applications (e.g., VRP).

- Weaknesses: Highly technical, assuming deep LR and statistical learning knowledge; no empirical validation; unclear extension to complex MILP instances.

---

> ### Author Rebuttal · Authors · 2026-03-31
>
> We thank the reviewer for the positive feedback. We are glad that the reviewer found our settings sound and our contributions novel. Also, in the revised draft, we additionally provide new results, which are: (1) closed the gap $\sqrt{s}$ in the rate of expected excess risk, (2) drawing a simple connection with learning with a parametric family, and (3) removing the uniqueness assumption on the solution of the LP problem in the learning-to-initialize. We hope that our new content improves the reviewer’s confidence in our draft, and we are happy to answer the reviewer’s concerns, if any. Many thanks!

---

> > ### Author Rebuttal · Reviewer_haky · 2026-04-04
> >
> > I am not familiar with this area, so I suggest the authors interact more with other reviewers.

---

### Decision · Program_Chairs · 2026-04-30

**Decision:**

Accept (regular)

**Comment:**

The authors consider a data-driven Lagrangian relaxation framework for Mixed-Integer Linear Programs (MILPs) and provide a statistical learning-like framework including generalization bounds and guarantees. The reviewers liked the theoretical results. The discussion was good and reviewers increased their scores.